# Abl family tyrosine kinases govern IgG extravasation in the skin in a murine pemphigus model

Sachiko Ono[1], Gyohei Egawa[1], Takashi Nomura[1], Akihiko Kitoh[1], Teruki Dainichi [1], Atsushi Otsuka[1], Saeko Nakajima[1], Masayuki Amagai[2], Fumi Matsumoto[3], Mami Yamamoto [3], Yoshiaki Kubota[4], Toshiyuki Takai[5], Tetsuya Honda[1] & Kenji Kabashima [1,6]

The pathway of homeostatic IgG extravasation is not fully understood, in spite of its importance for the maintenance of host immunity, the management of autoantibody-mediated disorders, and the use of antibody-based biologics. Here we show in a murine model of pemphigus, a prototypic cutaneous autoantibody-mediated disorder, that blood-circulating IgG extravasates into the skin in a time- and dose-dependent manner under homeostatic conditions. This IgG extravasation is unaffected by depletion of Fcγ receptors, but is largely attenuated by specific ablation of dynamin-dependent endocytic vesicle formation in blood endothelial cells (BECs). Among dynamin-dependent endocytic vesicles, IgG co-localizes well with caveolae in cultured BECs. An Abl family tyrosine kinase inhibitor imatinib, which reduces caveolae-mediated endocytosis, impairs IgG extravasation in the skin and attenuates the murine pemphigus manifestations. Our study highlights the kinetics of IgG extravasation in vivo, which might be a clue to understand the pathological mechanism of autoantibody-mediated autoimmune disorders.

[1] Department of Dermatology, Kyoto University Graduate School of Medicine, Kyoto, Japan. [2] Department of Dermatology, Keio University Graduate School of Medicine, Tokyo, Japan. [3] Research Unit/Immunology & Inflammation, Sohyaku, Innovative Research Division, Mitsubishi Tanabe Pharma Corporation, Yokohama, Japan. [4] Department of Anatomy, Keio University School of Medicine, Tokyo, Japan. [5] Department of Experimental Immunology, Institute of Development, Aging and Cancer, Tohoku University, Sendai, Japan. [6] Singapore Immunology Network (SIgN) and Skin Research Institute of Singapore (SRIS), Agency for Science, Technology and Research (A*STAR), Biopolis, Singapore. Correspondence and requests for materials should be addressed to G.E. (email: gyohei@kuhp.kyoto-u.ac.jp) or to K.K. (email: kaba@kuhp.kyoto-u.ac.jp)

mmunoglobulins (Igs), including autoantibodies, are initially produced in the lymphoid organs and circulate throughout the body via the blood. Considering the increased risk of cutaneous bacterial infections in patients with primary agammaglobulinemia[1], extravasation of Igs should exert essential protective immune functions especially in the skin. In addition, extravasation of Igs in the skin might also be related to the establishment of pathogenic immune reactions, such as cutaneous urticaria mediated by IgEs or the occurrence of autoimmune skin blistering disorders by autoantibodies against epidermal adhesion molecules[2–4].

It was reported that blood-circulating IgE can bind to the high-affinity Fcε receptors of dermal perivascular mast cells, which elongate their dendrites through the blood vessel wall[5]. As for IgG, although we recently showed that local inflammation enhances its extravasation[6], the mechanisms of homeostatic IgG extravasation was left unaddressed. Understanding the molecular mechanism of this process is essential for the management of autoantibody-mediated disorders. Moreover, considering that antibody-based biologics or immune checkpoint inhibitors are globally accepted as strong tools for the treatment of autoimmune disorders and various cancers, this insight into the basic peripheral kinetics of IgG would serve as practical information in the clinic.

Generally, there are two pathways of transvascular transport for plasma contents: paracellular (passive transport) and transcellular (active transport)[6]. Most blood vessels in peripheral tissues including the skin are composed of continuous blood vessels, and serve as a firm physiological barrier against plasma contents. Since the barrier prevents paracellular diffusion of albumin (Molecular size = 66 kilodalton [kDa]) and dextran with a molecular size of over 70 kDa[7–9], it seems unlikely that IgG (150 kDa) takes the paracellular pathway under homeostatic conditions. Nevertheless, a variety of plasma macromolecules takes the transcellular pathway by means of receptors expressed in vesicles of blood endothelial cells (BECs), which are derived from various forms of endocytosis[10–13]. Previous observations revealed the close localization of IgG and Fcγ receptors in vesicles, such as neonatal Fc receptor (FcRn) or FcγRIIb[14–18]. In addition, FcRn-unbound IgG is shown to enter into lysosomal degradation in BECs[19–23]. Although these studies implicate the importance of FcRn, Fcγ receptors, and endocytic vesicles for subcellular trafficking of IgG, their actual contribution to IgG extravasation in vivo remains to be addressed.

In relation to the transvascular transport, it was reported that a multityrosine kinase inhibitor, imatinib, relieved tissue edema upon inflammation in murine models of acute lung injury and cerebral infarction[24–27]. Another study also pointed out that imatinib relieves IgE-mediated anaphylactic reaction in mice[28]. The efficacy of imatinib in these models is mainly attributed to the decreased paracellular permeability via inhibition of the Abl family of tyrosine kinases[25,26,28]. In addition, imatinib and another Abl inhibitor was shown to downregulate caveolae-mediated endocytosis in vitro in BECs[29,30]. We thus hypothesized that imatinib may also modulate IgG extravasation in the skin under both inflammatory and homeostatic conditions.

Pemphigus vulgaris is a prototypic antibody-mediated disorder that targets a keratinocyte adhesion molecule, desmoglein (Dsg) 3, among other targets. Herein, using a murine model of pemphigus by targeting Dsg3, we examined how extravasation of blood-circulating IgG is regulated under homeostatic conditions. With the passive murine model used in this study, we did not address initiation of autoantibody response or the target specificity, but simply focused on the kinetics of IgG extravasation in vivo. In addition, we evaluated the relevance of Fcγ receptors

and the effect of imatinib and other Abl inhibitors on IgG extravasation in the skin.

## Results

**Blood-circulating IgG extravasates and reaches the epidermis.** We employed a passive murine pemphigus model that can be induced by intravenous injection of antimurine Dsg3 antibodies (Abs) to mice[31]. Among various antimurine Dsg3 Ab clones, a pathogenic clone (AK23) disrupts cell–cell adhesion between keratinocytes and induces hair loss in mice, but a nonpathogenic clone (AK18) does not[31] (Fig. 1a). Because the dissociation of keratinocytes may evoke local inflammation that may hinder the subsequent analysis, we used AK18 to evaluate the kinetics of IgG extravasation in a sensitive and quantitative way.

AK18, which belongs to the IgG1 subtype, was injected intravenously from the tail vein to mice. Twenty-four hours later, the deposition of AK18 in the ear epidermis was quantified by measuring the mean fluorescence intensity (MFI) of IgG1 in the $CD45^-E$-cadherin$^+$ keratinocytes by flow cytometry (hereafter referred to as IgG1-MFI) (Fig. 1b). IgG1-MFI started to elevate within 1 h and reached a plateau at 6 h after the intravenous injection (Fig. 1c). The injection doses of AK18 were associated with dose-dependent increases in IgG1-MFI ($r = 0.99$; evaluated 24 h after the AK18 injection) (Fig. 1d). To reflect high IgG1-MFI, AK18 deposition in the epidermis was detectable by immunohistochemistry when mice received a high dose (>100 μg) of Ab as we previously reported[6] (Fig. 1e). With injection of AK23, the manifestation of hair loss and epidermal AK23 deposition in host mice were detected only under a high-injection dosage (Fig. 1a, e). These results emphasize similar kinetics of AK18 and AK23, and support the idea that the amount of autoantibody deposition is critical for the establishment of the pemphigus skin manifestation. In order to check whether nonspecific IgG, not restricted to anti-Dsg3 Abs, extravasates from the blood to the dermal interstitium under homeostatic conditions, we intravenously injected fluorescein-conjugated nonspecific mouse IgG to mice. Twenty-four hours later, by the whole mount immunohistochemistry of the split ear skin, we found that dermal macrophages were labeled, suggesting that they received extravasated IgG in the dermis (Fig. 1f, g).

These results confirmed that the blood-circulating IgG, including autoantibodies, crosses over the blood vessel walls and distributes in the skin in a time- and circulating Ab dose-dependent manner under homeostatic conditions.

**Homeostatic IgG extravasation occurs transcellularly via BECs.** To determine whether IgG extravasation takes place via the transcellular pathway or the paracellular pathway under homeostatic conditions, we examined IgG kinetics by intravital-imaging analysis of murine ears. We intravenously injected fluorescein-conjugated IgG and found that injected IgG retained in the blood circulation. The leakage of IgG to the dermal interstitium was not apparent for an hour (Fig. 2a). Besides, the leakage of IgG was clearly demonstrated when local inflammation was induced by topical phorbol myristate acetate (PMA) to the ears (Fig. 2a and Supplementary Fig. 1A). On the other hand, internalization of AK18 in dermal BECs was detected by intracellular staining of IgG1 in the dermal BECs after an intravenous Ab injection (Fig. 2b, c). Therefore, the transcellular pathway is likely the main route for homeostatic IgG extravasation in the skin.

In the transcellular pathway, three major conventional ways are known for the formation of endocytic vesicles: caveolae-mediated endocytosis, clathrin-mediated endocytosis, and macropinocytosis[10]. Among them, dynamin governs endocytic membrane

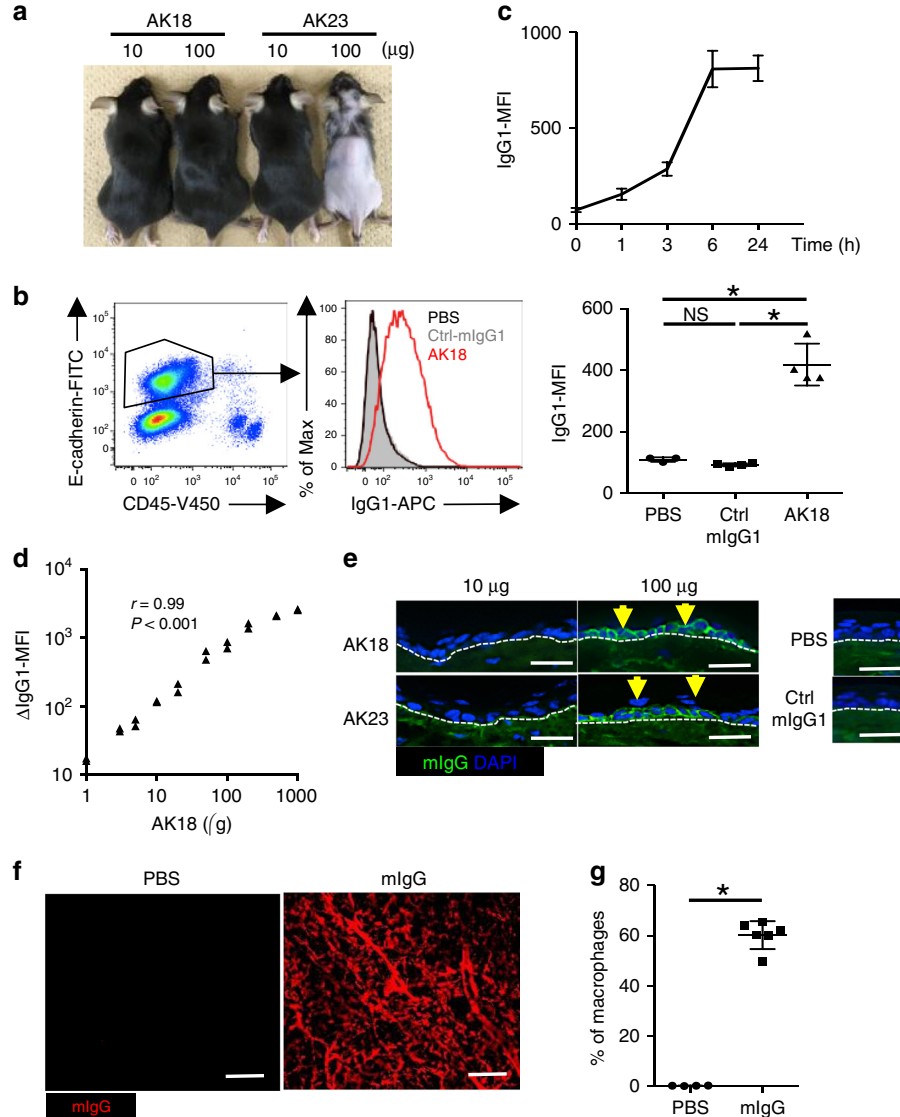

**Fig. 1** Kinetics of homeostatic IgG extravasation in the skin. **a** Manifestation of hair loss in mice 10 days after intravenous injection of AK18 or AK23 at 10 or 100 μg. **b** A representative flow cytometry plot and a histogram (the left and middle panels) to detect epidermal AK18 deposition in CD45+ E-cadherin+ keratinocytes, and IgG1-MFI (the right panel) in keratinocytes 24 h after intravenous injection of 20 μg AK18 ($n = 4$), 20 μg control mouse IgG1 (Ctrl-mIgG1) ($n = 4$), or PBS ($n = 3$). *$P < 0.05$ (a one-way ANOVA test). NS not significant. **c** A plot of epidermal IgG1-MFI against time after 100 μg of AK18 injection ($n = 2$ for each time point). **d** A plot of AK18 injection dose against epidermal ΔIgG1-MFI (the changes in IgG1-MFI from PBS-injected sample) 24 h after AK18 injection ($n = 2$ for each injection dose, two-tailed Spearman rank-correlation test). **e** Immunohistochemical evaluation in the mouse ear 24 h after intravenous injection of AK18 or AK23 at 10 or 100 μg, PBS, and 100 μg of control mouse IgG1 (Ctrl-mIgG1). Yellow arrows represent deposition of mouse IgG (mIgG) in keratinocytes. Blue represents nuclei stained with DAPI. White dotted line represents the border between the epidermis or the hair bulb and the dermis. Scale bar = 20 μm. **f** Images of ear skin dermis 24 h after intravenous injection of A555-conjugated mIgG or PBS. Scale bar = 100 μm. **g** The percentage of dermal macrophages positive for the fluorescein evaluated by flow cytometry, 24 h after intravenous injection of A647-conjugated mIgG ($n = 6$) or PBS ($n = 4$). *$P < 0.05$ ($t$-test). In each figure, the error bars represent the standard deviation of a data set

fission and formation of caveolae-mediated and clathrin-mediated endocytic vesicles. To evaluate the role of dynamin-dependent vesicles in BECs for IgG extravasation, we mated VE-cadherin-CreERT2 transgenic mice with dynamin 1 and 2 double conditional knockout mice (VE-cadherin-CreERT2; D1D2-floxed mice)[32,33]. In these mice, the formation of both clathrin-mediated and caveolae-mediated endocytic vesicles was ablated selectively in BECs after the intraperitoneal injections of tamoxifen. We confirmed by electron microscopy that vesicles in dermal BECs were diminished in Cre+ mice after tamoxifen treatment compared with Cre− control mice (Fig. 2d, red arrowheads). The interendothelial adhesion junction was unaffected in Cre+ mice as well as Cre− control mice under homeostatic condition

(Fig. 2d, yellow arrows). In such condition, both epidermal AK18 deposition (evaluated by IgG1-MFI) and internalization of AK18 in dermal BECs were reduced (Fig. 2e, f). Consistently, the enhancement of dermal macrophages after intravenous injection of fluorescein-conjugated IgG was also attenuated in Cre+ mice, and IgG retained in the blood circulation (Supplementary Fig. 1B). These observations indicate that the transcellular pathway is responsible for homeostatic IgG extravasation to the skin, and that dynamin-dependent endocytic vesicles in BECs are favorable cargo beds.

On the other hand, upon inflammation with topical PMA application to the ears, the ear swelling levels was unaffected (Fig. 2g), and the leakages of IgG to the interstitium were

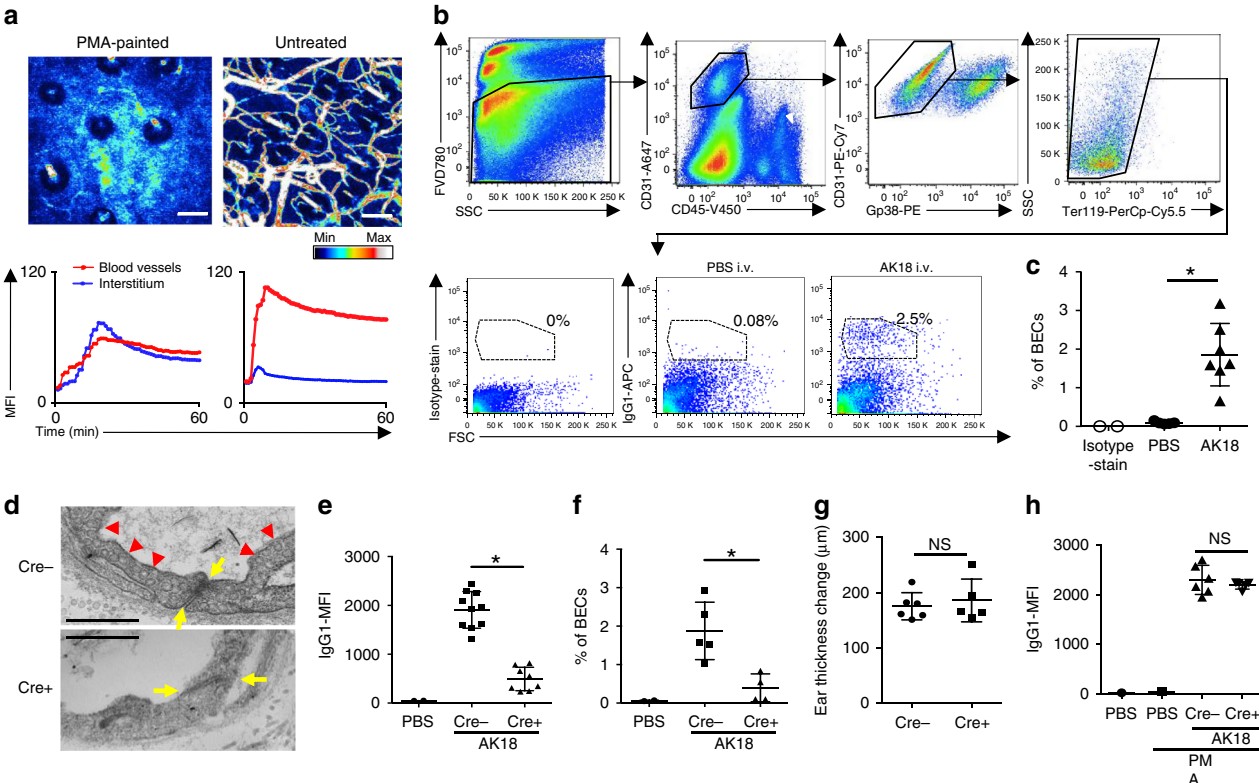

**Fig. 2** Homeostatic IgG extravasation in the skin relies on the transcellular pathway and dynamin-dependent endocytic vesicles in BECs. **a** Time-lapse images of paracellular IgG leakage (15 min after intravenous injection of fluorescein-conjugated IgG via the tail vein) in the ear dermis (the upper panels) exhibited in the rainbow color scale. Scale bar = 150 μm. Kinetics of MFI in the blood vessel area (red) and in the interstitium area (blue) in the dermis are plotted in the lower panels. **b** Representative flow cytometry plots for the identification of injected AK18 (dotted-line box) in dermal BECs of the mouse ear by flow cytometry. **c** The % frequency of IgG1-positive BECs 24 h after intravenous AK18 injection ($n = 7$) or PBS ($n = 5$), compared with isotype-stained control. *$P < 0.05$ ($t$-test). **d** Electron microscopic images of dermal BECs in the ear skin evaluated in VE-cadherin-CreERT2; D1D2-floxed mice in homeostatic condition. Red arrowheads show intracellular vesicles. Yellow arrows show interendothelial junctions. Scale bar = 500 nm. **e, f** Epidermal IgG1-MFI (**e**) and the % frequency of IgG1-positive BECs (**f**) 24 h after 100 μg of intravenous AK18 injection or PBS in VE-cadherin-CreERT2; D1D2-floxed mice ($n = 5$ for Cre$^-$ or $n = 4$ for Cre$^+$, both ears were evaluated separately in **e**). *$P < 0.05$ ($t$-test). **g, h** The ear thickness changes (**g**) and epidermal IgG1-MFI (**h**), 24 h after 20 μg of intravenous AK18 injection or PBS and topical PMA application to the ears, in VE-cadherin-CreERT2; D1D2-floxed mice ($n = 6$ for Cre$^-$ and $n = 5$ for Cre$^+$). In each figure, the error bars represent the standard deviation of a data set

comparable between Cre$^+$ and Cre$^-$ control mice (Supplementary Fig. 1C), after the intravenous injection of fluorescein-conjugated IgG. In addition, epidermal AK18 deposition of Cre$^+$ mice was also comparable with that of control mice in inflamed ears (Fig. 2h). These results suggest that the robust increase in the paracellular permeability upon inflammation is unaffected in VE-cadherin-CreERT2; D1D2-floxed mice, and that the defect of transcellular IgG extravasation is negligible in such situations. Together with our previous report showing that epidermal IgG deposition was markedly increased in inflamed ears compared with untreated ears[6], we concluded that the transcellular IgG-extravasation pathway is less important under inflammatory conditions when paracellular leakage is dominant.

**Caveolae mediates IgG extravasation in the skin.** We attempted to clarify which vesicles are responsible for homeostatic IgG extravasation. We cultured human dermal BECs (HDBECs) with fluorescein-conjugated human IgG and observed subcellular IgG distribution in HDBECs, which might be derived from active endocytosis. IgG in HDBECs colocalized with EEA1, a marker of early endosomes, in foci within the cytoplasm (stained with CellMask Green), while being resistant to the acid wash by acetic acid[34], suggesting that IgG was internalized in HDBECs (Fig. 3a and Supplementary Fig. 2). We found that this IgG endocytosis

by HDBECs was decreased when cultured with dynasore, an inhibitor of dynamin (Fig. 3b). The result was consistent with the observations in VE-cadherin-CreERT2; D1D2-floxed mice (Fig. 2e, f). To compare the dominance of clathrin- or caveolae-mediated endocytosis in IgG uptake, we conducted an immunocytochemical analysis. IgG taken up by HDBECs was circumscribed by or colocalized with caveolin 1, the principal protein of caveolae, rather than with clathrin (Fig. 3c). In addition, IgG endocytosis was suppressed in HDBECs with a caveolae-conformation inhibitor, nystatin[35] (Fig. 3d), or siRNA against *caveolin 1* but not siRNA against *clathrin* (Fig. 3e and Supplementary Fig. 3).

To study the effect of nystatin in vivo, we injected nystatin intraperitoneally before the intravenous AK18 administration, and evaluated epidermal IgG1-MFI. Correspondingly, we found that nystatin pretreatment significantly reduced epidermal AK18 deposition in mice (Fig. 3f). These results suggest the dominant contribution of caveolae-mediated endocytosis in IgG extravasation.

**Fcγ receptors play negligible roles in IgG extravasation.** As mentioned, many plasma macromolecules use their receptors expressed on the membrane of endocytic vesicles in the transcellular pathway. To examine whether IgG extravasation in the

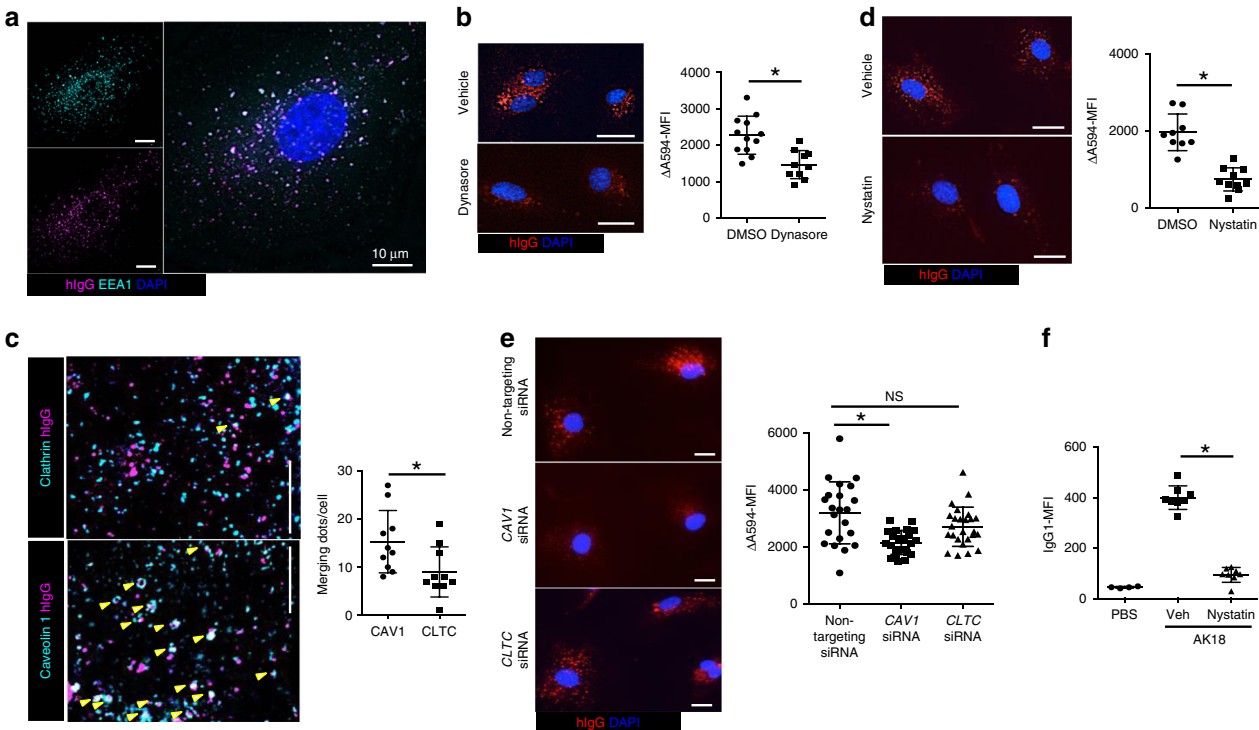

**Fig. 3** Caveolae mediate IgG endocytosis in BECs. **a** Subcellular human IgG (hIgG) distribution in BECs (magenda) co-stained with anti-EEA1 Ab (cyan). Blue represents a nucleus stained with DAPI. Scale bar = 10 μm. **b** hIgG distribution in dynasore- or vehicle (DMSO)-treated BECs. Scale bar = 20 μm. ΔA594-MFI of cells of each group is plotted in the right panel (n = 10 and 12 cells, each). *$P < 0.05$ (t-test). **c** Immunohistochemistry of caveolin 1 (CAV1) or clathrin (CLTC) (cyan) and hIgG (magenda) in BECs (the left panels). Yellow arrowheads represent the merging dots of green and red. Scale bar = 10 μm. The right panel shows the number of merging dots in each cell (n = 10 cells, each). *$P < 0.05$ (t-test). **d, e** The left panels show hIgG distribution in nystatin- or vehicle-treated BECs (**d**), and BECs treated with siRNA that targets caveolin 1 (CAV1), clathrin (CLTC), or nontargeting siRNA (**e**). Scale bar = 20 μm. The right panel shows ΔA594-MFI of BECs treated with nystatin or vehicle (DMSO) (**d**, n = 9 and 10 cells, each), and BECs treated with siRNA that targets caveolin 1 (CAV1), clathrin (CLTC), or nontargeting siRNA (**e**, n = 22, 23, and 25 cells, respectively) after the IgG endocytosis assay. *$P < 0.05$ (**d** t-test and **e** a one-way ANOVA test). **f** IgG1-MFI of ear epidermis 6 h after PBS (n = 4) or 100 μg of AK18 injection, pretreated with nystatin or vehicle (Veh) (n = 4, both ears were evaluated separately in each group). *$P < 0.05$ (t-test). In each figure, the error bars represent the standard deviation of a data set

skin also depends on any of Fcγ receptors, especially FcγRIIb, or FcRn, we examined the expression of Fcγ receptors or FcRn on/in murine dermal BECs. Similar to the previous observations of placental BECs[14,18], dermal BECs expressed low levels of FcγRIIb and/or FcγRIII and high levels of FcRn, but did not express FcγRI or FcγRIV (Fig. 4a).

To evaluate the role of FcγRIIb or FcγRIII for IgG extravasation in an in vivo setting, epidermal AK18 deposition was examined after intravenous Ab injection in FcγRIIb-deficient mice[36] and FcRg-deficient mice that lack the expression of Fcγ receptors other than FcγRIIb[37]. In both mouse strains, IgG1-MFI was found to be comparable with that of wild-type mice (Fig. 4b, c).

As for FcRn, we further employed β2-microglobulin (β2m)-deficient mice that lack functional FcRn[38]. The reliability of FcRn deficiency was first determined by evaluating transplacental AK18 transport to the fetal epidermis[39]. In brief, the β2m-hemi-deficient female mouse was mated with the β2m-hemi-deficient male mouse. Twenty-four hours after the intravenous administration of AK18 to the pregnant female mouse at gestational day 18.5, fetuses were delivered by cesarean section, and AK18 deposition in the fetal epidermis was analyzed by flow cytometry. Each fetal genotype was also examined. Supportively, AK18 deposition in the skin was completely blocked in β2m-deficient fetuses and half-reduced in β2m-hemi-deficient fetuses compared with wild-type littermates (Fig. 4d). In contrast, in adult mice, epidermal AK18 deposition and serum AK18 levels in β2m-deficient mice were comparable with those in control mice 6 h

after AK18 injection (Fig. 4e). These results suggest that FcRn is not essential for homeostatic IgG extravasation in the skin. Of note, epidermal AK18 deposition was decreased at a later time point (24 h after injection) in β2m-deficient mice (Fig. 4f, left), likely reflecting decreased serum AK18 levels caused by impaired IgG-recycling in BECs (Fig. 4f, right).

In addition, we applied intravenous immunoglobulin (IVIG) treatment as an alternative way to block FcRn function. Consistent with the previous report[39], IVIG at the dosage of 1.0 g kg$^{-1}$ day$^{-1}$ could completely prevent the transplacental AK18 transfer to fetuses (Fig. 4g). In this condition, however, epidermal AK18 deposition in the adult mice was not prevented: epidermal AK18 deposition and the serum AK18 levels were comparable with that of control mice at 6 h after AK18 administration (Fig. 4h), and both were partially decreased at 24 h after AK18 administration (Fig. 4i). These results were compatible with the results of β2m-deficient mice (Fig. 4d–f), and also supported our idea that FcRn is unessential for transvascular IgG transport in the skin.

Considering these observations, we concluded that the extravasation of IgG in the skin is mediated without the support of Fcγ receptors, including FcγRIIb or FcRn.

**Imatinib reduces IgG extravasation in the skin.** We next examined whether a multityrosine kinase inhibitor, imatinib, affects IgG extravasation, based on the increasing number of

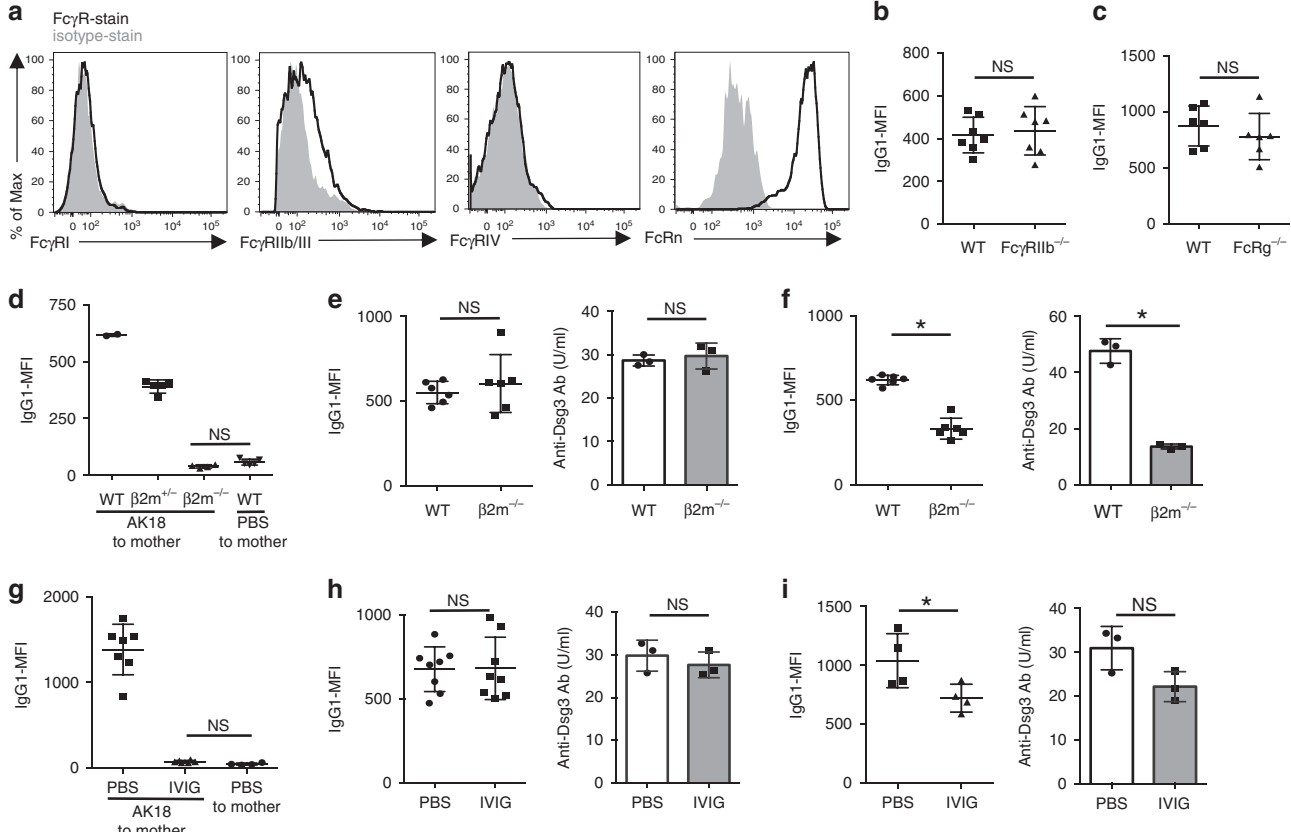

**Fig. 4** Dispensable contribution of Fcγ receptors for homeostatic IgG extravasation in the skin. **a** Expression of each Fcγ receptor on BECs in the ear skin (solid line). Fill drawing represents isotype-stained control. **b, c** IgG-MFI of ear epidermis 24 h after intravenous AK18 injection in FcγRIIb-deficient (FcγRIIb$^{-/-}$) mice (**b**) and FcRg-deficient (FcRg $^{-/-}$) mice (**c**), compared with wild-type (WT) mice ($n = 7$, each in **b** and $n = 6$, each in **c**). **d** IgG1-MFI of dorsal epidermis of neonates (WT [β2m$^{+/+}$], β2m-hemi-deficient [β2m$^{+/-}$], or β2m-deficient [β2m$^{-/-}$]) delivered from β2m$^{+/-}$ parents ($n = 2$, 5, or 4 for each littermate), 24 h after intravenous administration of AK18 or PBS ($n = 5$) to the mothers. **e** IgG1-MFI of ear epidermis (the left panel) or serum AK18 levels (the right panel) in WT or β2m$^{-/-}$ adult mice 6 h after intravenous AK18 injection ($n = 3$, each. Both ears were evaluated separately for IgG1-MFI). **f** IgG1-MFI of ear epidermis (the left panel) or serum AK18 levels (the right panel) in WT or β2m$^{-/-}$ adult mice 24 h after AK18 injection ($n = 3$, each. Both ears were evaluated separately for IgG1-MFI). *$P < 0.05$. **g** IgG1-MFI of fetuses from pregnant mice after intravenous AK18 injection, pretreated with 1.0 g kg$^{-1}$ day$^{-1}$ of IVIG or PBS ($n = 7$ or 6, each). PBS-injected mothers were used as the control ($n = 4$). **h** IgG1-MFI of ear epidermis (the left panel, $n = 4$, each. Both ears were evaluated separately) or serum AK18 levels (the right panel, $n = 3$) in adult mice 6 h after intravenous AK18 injection, treated with IVIG at 1.0 g kg$^{-1}$ day$^{-1}$ or PBS. **i** IgG1-MFI of ear epidermis (the left panel, $n = 4$, each) or serum AK18 levels (the right panel, $n = 3$, each) in adult mice 24 h after intravenous AK18 injection, treated with IVIG at 1.0 g kg$^{-1}$ day$^{-1}$ or PBS. A parametric Student's $t$-test was applied to each experiment. In each figure, the error bars represent the standard deviation of a data set

reports showing a unique effect of imatinib on regulating vascular permeability[24,26,27]. These reports demonstrated the impact of imatinib under inflammatory conditions. As such, we first evaluated whether imatinib could reduce IgG extravasation in inflammation.

As we previously reported, the local skin inflammation induced by various stimuli including topical PMA significantly enhances autoantibody deposition in the epidermis in our murine pemphigus model[6]. We administered imatinib intraperitoneally to mice before the intravenous AK18 injection and found that imatinib reduced epidermal AK18 deposition in the inflamed ears (Fig. 5a). Therefore, we assumed that imatinib regulated, at least in part, the paracellular permeability of IgG in our model. This observation is consistent with previous reports exhibiting the impact of imatinib on vascular permeability under inflammatory conditions. The PMA-induced ear swellings were unaffected by imatinib (Fig. 5b), probably due to the fact that the amount of ear swelling is determined not only by vascular permeability.

Next, we examined the influence of imatinib on IgG extravasation under homeostatic conditions. We treated mice

with imatinib or vehicle (distilled water: dH$_2$O) before the AK18 administration, and found that epidermal AK18 deposition was suppressed in a dose-dependent manner by imatinib (Fig. 5c). A high-dose imatinib pretreatment (6000 μg body$^{-1}$ day$^{-1}$) almost completely blocked Ab deposition, even when AK18 was injected at a high dose (up to 200 μg body$^{-1}$) (Fig. 5d). Serum AK18 levels were comparable between imatinib- and vehicle-pretreated mice (Fig. 5e), while internalization of AK18 in dermal BECs was largely decreased (Fig. 5f), similar to the results in VE-cadherin-CreERT2; D1D2-floxed mice (Fig. 2d, e). These results suggest that Ab extravasation is blocked at the phase of IgG endocytosis in BECs by imatinib.

We additionally tested if imatinib could also block extravasation of nonspecific IgG. Pretreatment of imatinib prevented the internalization of IgG by dermal macrophages after an intravenous injection of fluorescein-conjugated mouse IgG (Fig. 5g). This result suggests that IgG is constrained to the blood circulation by treatment with imatinib. Because this decrease in IgG uptake by dermal macrophages was recovered by an intradermal injection of IgG (Fig. 5g), it is unlikely that imatinib

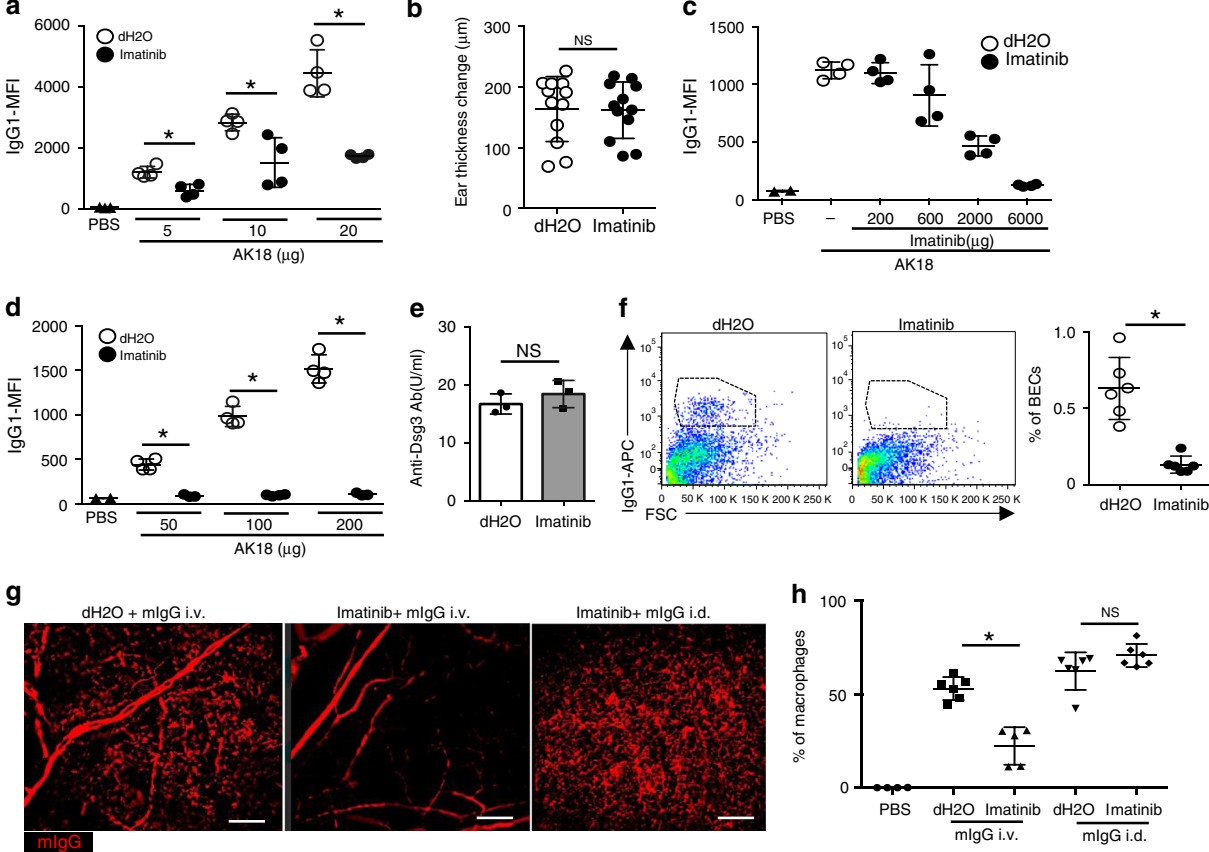

**Fig. 5** Imatinib reduces IgG extravasation in the skin. **a** IgG1-MFI of ear epidermis 24 h after 5–20 μg of AK18 injection and topical PMA to the ears, pretreated with imatinib (6000 μg body$^{-1}$) or vehicle (n = 4, respectively). *P < 0.05 (t-test). **b** The ear thickness changes 24 h after topical PMA to the ears pretreated imatinib (6000 μg body$^{-1}$) or vehicle (n = 6, each. Both ears were evaluated separately). **c** IgG1-MFI of ear epidermis 24 h after 100 μg of AK18 injection, pretreated with imatinib (0–6000 μg body$^{-1}$) or vehicle (n = 4, respectively). **d** IgG1-MFI of ear epidermis 24 h after 50–200 μg of AK18 injection pretreated with imatinib (6000 μg body$^{-1}$) (n = 3 or 4, respectively). *P < 0.05 (t-test). **e** Serum AK18 levels 24 h after 100 μg of AK18 injection pretreated with imatinib (6000 μg body$^{-1}$) or vehicle (n = 3, each). **f** The left panel shows IgG1-positive cells in BECs 24 h after 100 μg of AK18 injection with imatinib (6000 μg body$^{-1}$) or vehicle pretreatment. The right panel shows % frequency of IgG1-positive cells in BECs (n = 6). *P < 0.05 (t-test). **g**, **h** The whole mount image of the ear dermis (**g**) and the percentage of dermal macrophages evaluated by flow cytometry (**h**), positive for fluorescein after intravenous (i.v.) or intradermal (i.d.) fluorescein-conjugated mIgG injection, with imatinib (6000 μg body$^{-1}$) or vehicle pretreatment (n = 6 or 5 for i.v. and n = 6 for i.d., respectively). Scale bar = 100 μm *P < 0.05 (t-test). In each figure, the error bars represent the standard deviation of a data set

functions by inducing the dysfunction of phagocytic activity of macrophages. Flow cytometric analysis yielded similar results, showing that the imatinib pretreatment reduced the percentage of fluorescein-positive dermal macrophages after an intravenous injection of fluorescein-conjugated mouse IgG, but not after an intradermal injection (Fig. 5h).

These results suggest that imatinib blocks IgG extravasation to the skin not only upon inflammation but also under homeostatic conditions by interfering in both paracellular leakage of IgG as well as endocytosis of IgG in BECs.

**Abl family tyrosine kinases in BECs govern IgG extravasation.**
Imatinib is known to target the Abl family of nonreceptor tyrosine kinases (c-Abl and Arg) and some membrane-bound tyrosine kinase receptors such as platelet-derived growth factor receptor (Pdgfr)-α, Pdgfr-β, and c-Kit[40]. As mentioned, the efficacy of imatinib on vascular permeability seems to be mainly attributed to the inhibition of Abl family tyrosine kinases[25,26,28]. Of note, a previous report proposed the effect of imatinib on regulating vascular permeability during ischemic stroke through its action on Pdgfr-α on pericytes[27]. This raises the possibility that tyrosine kinases other than those of the Abl family may also play a role in IgG extravasation.

To define the importance of Abl family tyrosine kinases on IgG extravasation, we examined the contribution of other membrane-bound tyrosine kinases (Pdgfr-α, Pdgfr-β, and c-Kit) on BECs. We analyzed the expression of these tyrosine kinase receptors on murine dermal BECs by flow cytometric analysis, and found expression of Pdgfr-α, but not Pdgfr-β or c-Kit (Supplementary Fig. 4A). We then pretreated mice with a sufficient dose of an anti-Pdgfr-α blocking Ab, APA5[41], before the AK18 administration, however, epidermal AK18 deposition was not affected (Supplementary Fig. 4B).

We also examined the possible contribution of mast cells, since imatinib affects their functions by inhibiting c-Kit activity[42]. Dermal mast cells locate along blood vessels and were shown to have a potency to capture blood-circulating IgE via Fcε receptors[5]. Because mast cells also possess Fcγ receptors (FcγRIIb and/or FcγRIII but not FcγRI nor FcγRIV) (Supplementary Fig. 4C), we assumed that they might even be engaged in transferring IgG to tissue interstitium. Moreover, even under homeostatic conditions, subtle physiological stimuli may activate mast cells to induce histamine release and increase blood vessel permeability[6]. To reveal the role of mast cells in homeostatic IgG extravasation, we employed inducible mast cell-deficient (Mas-TRECK) mice. Even under mast cell depletion (Supplementary

Fig. 4D), IgG1-MFI was comparable with that in mast cell sufficient controls (Supplementary Fig. 4E).

To further investigate the direct effect of Abl family tyrosine kinases on IgG extravasation, we examined IgG endocytosis by HDBECs that were exposed to siRNA against *c-ABL*. We confirmed that IgG endocytosis by HDBECs was decreased when cultured with siRNA against *c-ABL*, as well as siRNA against *caveolin 1* (Fig. 6a). In addition, we conducted an in vitro transwell permeability assay using human dermal microvascular endothelial cells (HDMECs). As expected, imatinib showed an inhibitory effect on thrombin-induced IgG transition through HDMECs in a dose-dependent manner (Fig. 6b). Similarly, asciminib, another specific inhibitor of c-Abl tyrosine kinase[43], also showed reduced IgG transition through HDMECs (Fig. 6c). Inhibition of both Abl and Arg with GNF-2, which displays a greater target specificity than imatinib and is not known to inhibit any additional kinases[44,45], almost completely abolished IgG transition through HDMECs at concentrations >1 μM (Fig. 6d). These results indicate that Abl family tyrosine kinases, including both c-Abl and Arg, are essential for IgG transport through BECs.

We additionally evaluated the in vivo effect of reagents that have a specific effect on Abl family tyrosine kinases other than imatinib. We examined 5-aminoimidazole-4-carboxamide ribonucleotide (AICAR)[30], which was newly identified to inhibit Abl phosphorylation by activating adenosine monophosphate-dependent kinase. Although increasing number of studies assessed the antimetabolic and immunomodulatory effect of AICAR and its analog in vivo[46–48], its effect on IgG kinetics remains unknown. Pretreatment with AICAR was effective in reducing epidermal AK18 deposition (Fig. 6e). These results support that Abl family tyrosine kinases in BECs play an essential role in homeostatic IgG extravasation in the skin.

**Imatinib reduces extravasation of IgG but not of dextran**. To examine if the effect of imatinib is specific on caveolae and extravasation of IgG, mice were intravenously injected with Alexa-Fluor 594 (A594)-conjugated mouse IgG together with fluorescein isothiocyanate (FITC)-conjugated 150 kDa dextran (a similar molecular weight with IgG), which is generally considered to be taken up by macropinocytosis and/or noncaveolae and clathrin-independent endocytosis[49–52]. Extravasation of both molecules was depicted as enhancement of dermal macrophages that engulfed these macromolecules (Supplementary Fig. 5A). While imatinib reduced extravasation of IgG, it did not reduce the extravasation of 150 kDa dextran (Supplementary Fig. 5A). Likewise, flow cytometric analysis demonstrated that the percentage of dermal macrophages that engulfed FITC-conjugated dextran was not decreased by the imatinib pretreatment (Supplementary Fig. 5B). These results indicate the different extravasation routes of IgG and dextran, and suggest a selective effect of imatinib on caveolae-mediated endocytosis in BECs.

**Imatinib prevents the hair loss in a murine pemphigus model**. We tested the possibility of regulating IgG extravasation as a therapeutic target for autoantibody-mediated disorders. Mice were treated with imatinib before the administration of a pathogenic autoantibody, AK23. Imatinib attenuated the development of severe hair loss in mice (Fig. 7a) by preventing hair acantholysis (the disruption of cell–cell adhesion between keratinocytes in the hair follicles) (Fig. 7b). AK23 deposition in interfollicular and follicular keratinocytes was ablated for 5 days after imatinib treatment (Fig. 7c). Similar to the observation for AK18 (Fig. 5e), serum AK23 levels were comparable between imatinib- and vehicle-treated mice until the elimination of injected AK23 from the blood (Fig. 7d), suggesting that imatinib

had a marginal effect on degradation or recycling of auto-antibodies. In contrast to intravenous injection, hair loss (Fig. 7e) and AK23 deposition in interfollicular and follicular keratinocytes (Fig. 7f) was not abrogated by imatinib after the intradermal injection of AK23. This supports the idea that imatinib is likely to elicit its action via blocking Ab extravasation rather than interfering with Ab deposition to the epidermis or attenuation of its pathogenic activity. These observations suggest that imatinib has a potential to prevent the development of murine pemphigus skin manifestations.

We additionally tested if imatinib could also block extravasation of autoantibodies in mice obtained from pemphigus patients, capitalizing on the ability of pemphigus autoantibodies in humans to react with murine Dsgs[6]. Mice were intravenously injected with sera obtained from two pemphigus patients, who are positive for anti-Dsg3 Abs composed of IgG1 and IgG4 subclasses. By immunohistochemical analysis, we detected both human IgG1 and IgG4 deposition in the epidermis when mice were injected with 100 μl of each patient's serum, whereas it became undetectable with imatinib pretreatment (Supplementary Fig. 6). Taken together, these findings allowed us to consider that imatinib reduces the autoantibody deposition in the skin and protects from the autoantibody-mediated pathological changes.

## Discussion

Our findings highlight a distinct pathway for homeostatic IgG extravasation in vivo and the impact of Abl family tyrosine kinases on IgG kinetics at the blood–tissue interface. Homeostatic IgG transport from the blood to the skin interstitium occurs via the transcellular pathway through dermal BECs, which uses IgG endocytosis by caveolae under the control of Abl family tyrosine kinases. Increasing number of studies pointed out the efficacy of inhibiting Abl family tyrosine kinases for regulating permeability of BECs in inflammatory conditions; which was, however, unknown for IgG kinetics and even for homeostatic conditions.

In contrast to the previous considerations[22,23], major Fcγ receptors seem to be dispensable for the IgG extravasation process. Although FcRn was mandatory for the transportation of IgG through the placenta (Fig. 4d) as we previously demonstrated[39], FcRn was dispensable for transportation of IgG through the blood vessels to the skin interstitium. Consistently, we showed that extravasation of human pemphigus autoantibodies (which belongs to either IgG1 or IgG4) to the skin was blocked by imatinib. It may also be possible that other subtypes of Igs share this transport mechanism, which requires further investigation. Our results also imply that there may be a distinct endocytic pathway for IgG, as opposed to dextran, that is generally considered as a marker of fluid-phase endocytosis[49]. Although it is considered that IgG is also taken up by fluid-phase endocytosis by BECs[22], there might be unknown factors or rules that govern the choice of endocytic vesicle as a cargo bed in BECs under various settings.

In our study, it is possible that Abl family tyrosine kinases or the depletion of Fcγ receptors might have also affected the function of immune cells in vivo. However, since the dissociation of cell–cell adhesion between keratinocytes can be induced solely by deposition of anti-Dsg3 Ab to keratinocytes without contribution of immune cells[31], we think that in our passive murine pemphigus model, we could evaluate the direct effect of Abl family tyrosine kinases on IgG kinetics. This insight into the basic IgG kinetics in vivo, as well as the efficacy of Abl family tyrosine kinases on IgG kinetics, may attract interest from both the immunological and the clinical fields. At the same time, with the increasing attention to Abl family tyrosine kinases as a tool for treating not only cancerous but also noncancerous

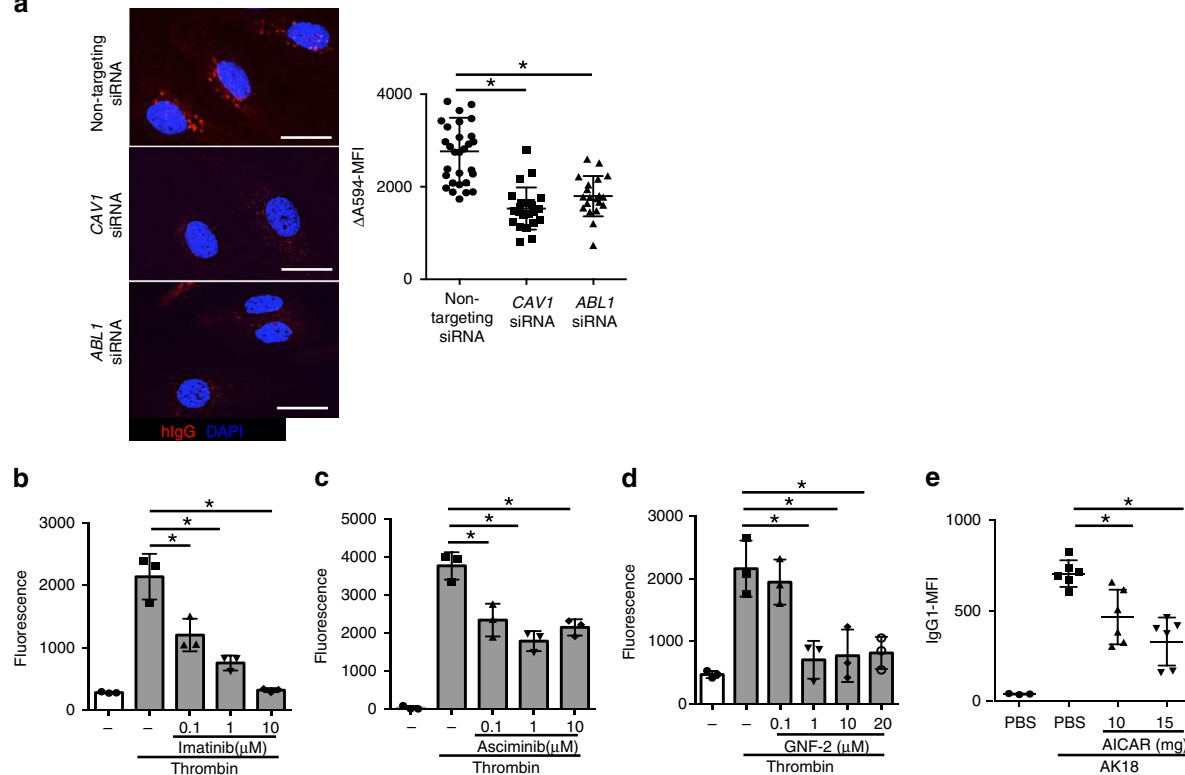

**Fig. 6** Abl family tyrosine kinase inhibitors reduce IgG transport through BECs. **a** The left panels show hIgG distribution in BECs treated with siRNA that targets *caveolin 1* (*CAV1*), *c-ABL* (*ABL1*), or nontargeting siRNA. Scale bar = 20 μm. The right panel shows ΔA594-MFI of BECs of each cell (*n* = 30, 22, and 20 cells, respectively), and BECs treated with siRNA that targets *caveolin 1*(*CAV1*), *c-ABL* (*ABL1*), or nontargeting siRNA (E, *n* = 22, 23, and 25 cells, respectively) after the IgG endocytosis assay. *$P < 0.05$ (a one-way ANOVA test). **b–d** IgG transition through HDMECs was evaluated by the in vitro transwell permeability assay under thrombin stimulation, treated either with imatinib at 0.1–10 μM (**b**), asciminib at 0.1–10 μM (**c**), or GNF-2 at 0.1–20 μM (**d**), or vehicle (*n* = 3 wells, each). *$P < 0.05$ (a one-way ANOVA test). **e** IgG1-MFI of ear epidermis 24 h after 100 μg of AK18 injection pretreated with vehicle or AICAR (*n* = 6, respectively). *$P < 0.05$ (a one-way ANOVA test). The precise protocol is described in the "Methods" section. In each figure, the error bars represent the standard deviation of a data set

diseases[26,28,42,46–48], we should also note that these reagents can be of aid in the dysregulation of host immunity by reducing IgG concentration in the peripheral tissues.

The extravasation of IgG should be beneficial for the host defense against infectious pathogens in peripheral tissues, and yet can also trigger autoantibody-mediated manifestations, as such demonstrated in our murine pemphigus model. Our observations were consistent with previous in vitro reports that indicate the importance of c-Abl on caveolae-mediated endocytosis[29,30]. We propose that the strategy to block Abl family tyrosine kinases, such as by imatinib or AICAR, might be a useful alternative in the treatment of autoantibody-mediated disorders. Although it is important to note that the efficacy of these drugs might vary among target tissues due to the differences in the rigidity of the intercellular barrier of blood capillaries, as well as the dependency on the transcellular pathway for IgG extravasation[7]. Autoantibody-mediated encephalitis or retinopathy can be other targets, considering the tight blood barrier structures in these tissues. On the other hand, considering the efficacy of imatinib on IgG extravasation upon inflammation (Fig. 5a), together with its effect on pathological vascular hyperpermeability[24–26], imatinib might also be useful even with disruption of the blood vessel barrier. We did not precisely exclude the effect of imatinib on the paracellular IgG diffusion in the skin under both homeostatic and inflammatory condition in this study, which would be another important issue to address in the future. Besides, we found that imatinib

had little effect on the transition of serum anti-Dsg3 Ab titers. Although the definite mechanism is still undiscovered, it seems to be consistent with the notion that BECs not only recycle but also degrade the internalized IgG[23,53]. We assume that both the degradation and recycling processes were inhibited by blocking the endocytosis of IgG, resulting in equivalent serum autoantibody titers.

Conversely, our study has several limitations and issues to be solved. First, it remains unsolved how Abl family tyrosine kinases are activated under homeostatic conditions, although previous reports implied that physiological oxidative stresses to BECs might be involved in this process[29,30]. Second, the contribution of other molecules such as c-Src kinase in regulating IgG extravasation needs to be clarified. Although it is considered that imatinib has low affinity to c-Src kinase[54], c-Src kinase is also known to govern caveolar function;[55] therefore, intervention of other unknown molecules might be undeniable. As mentioned, our most important query is whether Abl family tyrosine kinases can be applied to treat autoantibody-mediated disorders. On this, we are now undergoing a clinical trial of imatinib in patients with autoimmune-blistering diseases in Kyoto University Hospital (The study is registered with the UMIN Clinical Trials Registry, number UMIN000030865).

In conclusion, our study provided insights into the kinetics of IgG extravasation in vivo and revealed a previously unknown immunomodulatory aspect of inhibitors for Abl family tyrosine kinases, including imatinib, against autoantibody-mediated

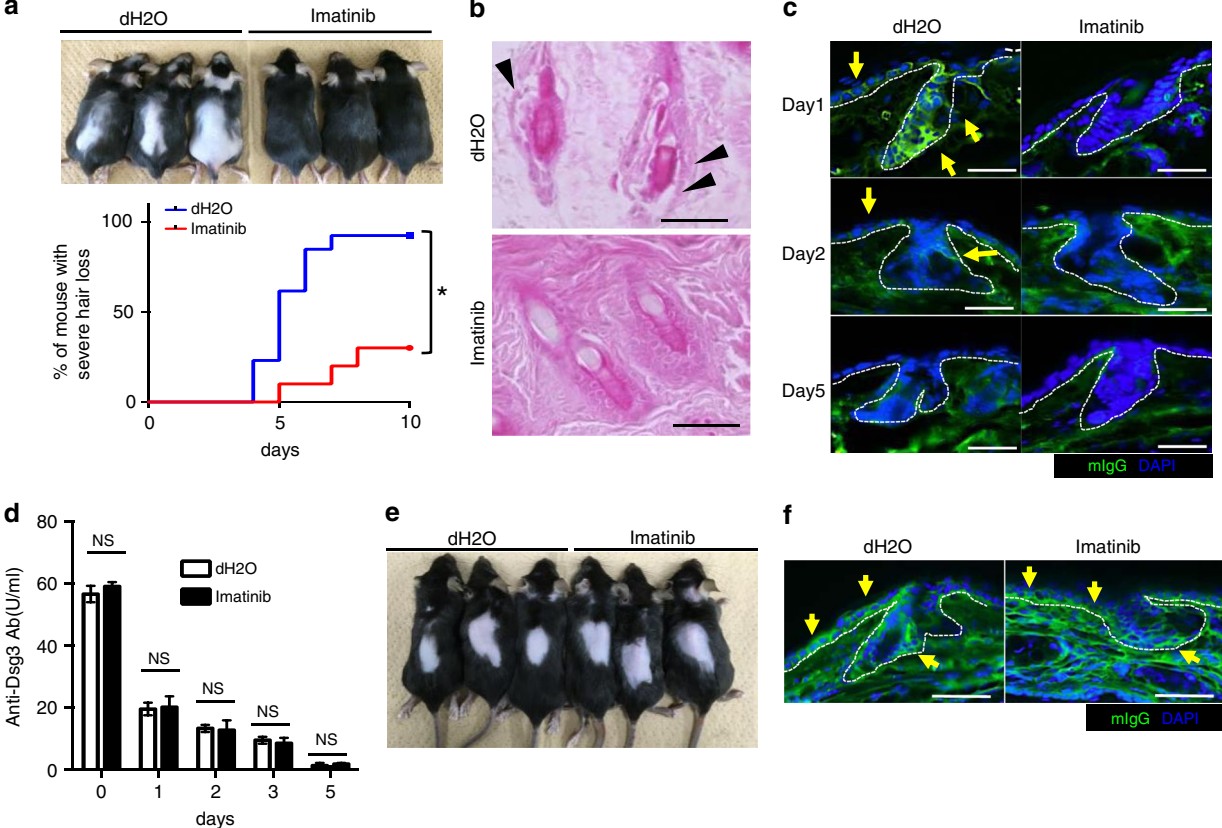

**Fig. 7** Imatinib prevents the development of murine pemphigus manifestations. **a** The upper panel shows manifestation of hair loss 10 days after intravenous injection of AK23, pretreated with imatinib or vehicle. The percentage of mice that developed severe hair loss (over 30% of their back skin area) was plotted in the lower panel ($n = 10$ or 13, respectively, in accumulation of three trials). *$P < 0.05$ (a log-rank test). **b** Histology of hair follicles in the back skin of mice 4 days after intravenous injection of AK23 pretreated with imatinib or vehicle. Arrowheads represent the disruption of cell–cell adhesion in follicular keratinocytes. Scale bar = 50 μm. **c** Immunohistochemical evaluation for AK23 deposition (green) in interfollicular and follicular keratinocytes in the ear skin of mice 1, 2, or 5 days after intravenous injection of AK23, pretreated with imatinib or vehicle. **d** Serum AK23 levels 24 h after intravenous Ab injection, pretreated with imatinib or vehicle ($n = 4$ for day 0–3, $n = 3$ for day 5). **e** The manifestation of hair loss after intradermal AK23 injection into the back skin of mice pretreated with imatinib or vehicle. **f** Immunohistochemical evaluation for AK23 deposition (green) in interfollicular and follicular keratinocytes in the ear skin of mice after intradermal AK23 injection into the ears, pretreated with imatinib or vehicle. In **c** and **f**, yellow arrows indicate intercellular AK23 deposition. Blue represents nuclei stained by DAPI. White dotted line represents the border between the epidermis or the hair bulb and the dermis. Scale bar = 50 μm. In each figure, the error bars represent the standard deviation of a data set

disorders. A better understanding of the IgG kinetics might serve as an asset in the search for therapeutics of autoantibody-mediated disorders, and also in host immunity.

## Methods

**Mice**. Female 8- to 10-week-old C57BL/6-background mice were used in this study unless otherwise indicated. The age of mice used in AK23 trials was between 9 and 10 weeks old. C57BL/6 and BALB/c mice were purchased from SLC (Shizuoka, Japan).

FcγRIIb-deficient mice[36] and FcRg-deficient mice[37] were gifted from Dr T. Takai, and both mouse strains were backcrossed with C57BL/6 mice for at least more than ten generations before the experiments. β2m-deficient mice were gifted from Dr Y. Hamasaki[56].

Mas-TRECK mice were as described previously[57], and to deplete mast cells in vivo, 250 ng of diphtheria toxin dissolved in 200 μl of PBS was intraperitoneally injected to Mas-TRECK mice for 5 consecutive days. The experiments were conducted the next day following the last injection of diphtheria toxin.

VE-cadherin-CreERT2 transgenic mice[33] were mated with dynamin 1 and 2 double conditional knockout mice[32] to establish VE-cadherin-CreERT2; D1D2-floxed mice. Both strains were kindly gifted from Dr Y. Kubota, and mice were backcrossed with C57BL/6 mice for at least more than ten generations before the mating. Two milligrams of tamoxifen (Sigma Aldrich, St. Louis, MO) was dissolved in 200 μl of ethanol/sunflower oil (1:8) and intraperitoneally injected to mice for 5 consecutive days. The experiments were conducted 6–7 days after the last injection of tamoxifen. Cre negative littermates were used as controls.

**Passive murine pemphigus model**. Hybridoma cells that produce pathogenic (AK23) and nonpathogenic (AK18) anti-mouse Dsg3 monoclonal antibodies were provided from Keio Unversity[31]. To obtain antibodies, these cells were cultured in GIT medium (Wako, Tokyo, Japan) at 37 °C for 14 days and IgG was purified from culture supernatant using a Protein G HiTrap column (GE Healthcare, Uppsala, Sweden). For flow cytometric analysis, 100 μg (to check IgG1-MFI in untreated ears) or 20 μg (to check IgG1-MFI in inflamed ears) of AK18 was intravenously administered to mice via the tail vein, and the ear skin was collected 24 h later unless otherwise indicated. For immunohistochemical analysis, 100 μg of AK23 was intravenously administered via the tail vein, and the ear skin was collected 1–5 days later. Otherwise, 50 or 5 μg of AK23 was intradermally administered to the back skin or the ear skin of mice, respectively. Hair loss started 4–5 days after AK23 injection, and progressed until 10 days after AK23 injection. The images of hair loss were obtained 10 days after AK23 injection. The serum levels of anti-mouse Dsg3 Ab were analyzed using an enzyme-linked immunosorbent assay (ELISA) kit (MBL, Nagoya, Japan) according to the manufacturer's instructions.

**Cell isolation and flow cytometry**. To obtain a single-cell suspension from the epidermis, the dorsal halves of mouse ears were floated on 5 mg ml$^{-1}$ dispase II (Godo Shusei Co., LTD, Tokyo, Japan) dissolved in RPMI 1640 (Invitrogen, Carlsbad, CA) containing 10% fetal calf serum (FCS) for 30 min at 37 °C. Subsequently, the epidermis was manually separated from the dermis by fine tweezers. The epidermis was then floated on 0.25% trypsin-EDTA for 8 min at 37 °C and filtered through a 70-μm nylon mesh. For the dermal cell suspension, the dermis was digested with 0.33 mg ml$^{-1}$ liberase TL (Roche, Basel, Switzerland) dissolved in RPMI 1640 containing 10% FCS for 60 min at 37 °C and filtered through a 70-μm nylon mesh. Brefeldin A (final concentration = 10 ng ml$^{-1}$; Sigma Aldrich) was

added to dispase II and liberase TL solution in order to evaluate the amount of intercellular AK18 in BECs.

To stain cells, anti-mouse CD45 (Cat. 560501, clone 30-F11), E-cadherin (Cat. 560061, clone 36/E-Cadherin), IgG1 (Cat. 550874, clone X56), CD11b (Cat. 560455, clone M1/70), Ter119 (Cat. 560512, clone TER-119), and MHC classII (Cat. 563163, clone M5/114) Abs were purchased from BD Bioscience (San Jose, CA). Anti-mouse CD31 (Cat. 102516, clone MEC13.3), gp38 (Cat. 127407, clone 8.1.1), F4/80 (Cat. 123114, clone BM8), FcγRI (Cat. 139303, clone X54-5/7.1), FcγRII/III (Cat. 101307, clone 93), FcγRIV (Cat.149503, clone 9E9), Pdgfr-α (Cat. 135905, clone APA5), and CD200R3 (Cat. 142206, clone Ba1.3) Abs were purchased from BioLegend (San Diego, CA). Anti-mouse c-Kit (Cat. 17-1171-83, clone 288), FcεRI (Cat. 11-5898-85, clone MAR-1), and Pdgfr-β (Cat. 12-1402-81, clone APB5) Abs were purchased from eBioscience (San Diego, CA). Anti-mouse FcRn Ab (Cat. AF6775) was purchased from R&D systems (Minneapolis, MN), and stained subsequently with anti-goat A647 secondary Ab (Invitrogen). Fixable viability dye (Cat. 65-0865-18, eBioscience) was used to exclude dead cells.

To check IgG1-MFI in the epidermis, epidermal cells were stained with antibodies against CD45, E-cadherin, and IgG1, then fixed with Cytofix/Cytoperm (Cat. 51–2090KZ, BD Bioscience), and underwent flow cytometric analysis. We gated CD45−E-cadherin+ fraction to exclude intraepidermal immune cells and cell-debris. To evaluate cell-internalized IgG1 in dermal BECs, dermal cells were stained with fixable viability dye, antibodies against CD31, gp38, Ter119, and CD45. Cells were then fixed and permeabilized with Cytofix/Cytoperm and Perm wash (Cat. 51-2091KZ, BD Bioscience), and underwent intracellular staining with anti-mouse IgG1 antibody for flow cytometric analysis.

To detect the extravasation of nonspecific IgG or dextran in the murine ears by flow cytometry, 100–200 μg of fluorescein (A647)-conjugated mouse IgG (Cat. 015-600-003, Jackson ImmunoResearch, West Grove, PA) and/or 1 mg of FITC-conjugated 150 kDa dextran (Sigma Aldrich) was intravenously injected via tail vein to mice or was intradermally injected to the ears of mice. The dermal cell suspension was collected after extraction of epidermis with dispase II, and subsequently digested with liberase TL, as mentioned above. Cells were stained with fixable viability dye, anti-mouse CD45, CD11c, CD11b, MHC classII, and F4/80. Cells were then fixed with Cytofix/Cytoperm for flow cytometric analysis.

Flow cytometric analysis was performed using LSR Fortessa (BD Bioscience) and analyzed by FlowJo software (Tree Star, Ashland, OR). The exact sequential gating strategy of dermal BECs is located in Fig. 2b. The gating strategies to check Ab deposition to epidermis and dermal macrophages are located as Supplementary Fig. 7A and 7B. Also, gating strategies of mast cells are located as Supplementary Fig. 7C.

**Time-lapse imaging of blood-circulating IgG by two-photon microscopy.** Female 8-week-old BALB/c mice were used for intravital-imaging analysis to avoid the influence of melanin. Mice were positioned on the heating-plate on the stage of a two-photon microscope, IX-81 (Olympus, Tokyo, Japan), and their ear lobes were fixed beneath cover slips with a single drop of immersion oil. Subsequently, 5 mg of human polyclonal IgG (Nihon Pharmaceutical Co. LTD, Tokyo, Japan) labeled with FITC (Sigma Aldrich) was injected via the tail vein. Stacks of 15 images, spaced 3.5 μm apart, were acquired every minute for 1 h. To evaluate in vivo IgG kinetics, the single color-scale images of fluorescein were obtained and converted to rainbow color-scale images according to fluorescent intensity using ImageJ soft-whare[58]. Then, the blood vessel area and the interstitial space were manually circumscribed[9], and the MFI of each area at each time point was calculated using ImageJ software. To induce skin inflammation, 20 μl of PMA (0.1 mg ml−1 in acetone; Sigma Aldrich) was painted on the ear 24 h before the IgG injection.

**Immunohistochemistry of the murine ear skin.** To detect AK23-induced hair acantholysis, a disruption of cell–cell adhesion between follicular keratinocytes, the murine dorsal skin samples were obtained 4 days after intravenous injection of AK23, fixed in 10% formaldehyde, and embedded in paraffin. Samples were then sliced at 5 μm, deparaffinized, and underwent hematoxylin and eosin staining.

To detect epidermal AK18 or AK23 deposition in the ear skin, the ear skin samples were obtained 24 h after intravenous injection of antibodies, fixed in 4% paraformaldehyde (PFA), and embedded in O.C.T. compound (Sakura Finetek, Torrance, CA). Five micrometers of slices were treated with Image-iT FX signal enhancer (Invitrogen), and incubated for 30 min at room temperature with goat anti-mouse IgG A594 secondary Ab (Invitrogen). PBS or control mouse IgG1 (Cat. 400166, clone MOPC-21, BioLegend) was injected as negative controls. The slices were mounted with ProLong Diamond Antifade Mountant with DAPI (Life Technologies, Carlsbad, CA).

To detect epidermal autoantibody deposition after injecting sera from pemphigus patients, sera from two pemphigus vulgaris patients (PV#1 and PV#2), who were positive for anti-Dsg3 Ab but not anti-Dsg1 Ab, were collected for the analysis. Among the IgG1-IgG4 subclasses, both sera were positive for the IgG1 and IgG4 subclasses. The ears of mice were collected 24 h after intravenous injection of each serum, then fixed, mounted, and stained with anti-human IgG FITC (Abcam, Cambridge, UK). With intravenous injection of each serum of PV#1 and PV#2 to adult mice via the tail vein, we determined that at least 100 μl of each patient's serum was required to obtain apparently positive results in the untreated ears. Due to the limitation of the serum amount, the experiments using patients'

sera were performed in a single mouse and its control, twice, with similar results for each serum.

To detect the extravasation of nonspecific IgG or dextran in the murine ears by immunohistochemistry, 100–200 μg of fluorescein (A555 or A594)-conjugated mouse IgG (Cat. 015-160-003 or 015-600-003, Jackson ImmunoResearch, West Grove, PA) and/or 1 mg of FITC-conjugated 150 kDa dextran (Sigma Aldrich) was intravenously injected via the tail vein to mice. Otherwise, 20 μg of fluorescein-conjugated mouse IgG was intradermally injected to the ears. The ears of mice were collected 24 h after injection and the dorsal halves of the ears were floated on 4% PFA at 4 °C for a day. The ears were mounted with ProLong Diamond Antifade Mountant without DAPI (Life Technologies, Carlsbad, CA).

These samples were observed under a fluorescent microscope, BZ-900 (KEYENCE, Osaka, Japan) or a confocal microscope (Nikon, Tokyo, Japan).

**Reagents and in vivo blocking assay.** To block caveolae-mediated endocytosis, mice were intraperitoneally treated with 8 mg kg−1 body weight of nystatin (Sigma Aldrich) 1 h before AK18 administration. Ear samples were evaluated for IgG1-MFI 6 h after AK18 administration. Nystatin was first dissolved in dimethyl sulfoxide (DMSO) at 25 mg ml−1, and diluted to the final concentration of 1 mg ml−1 in distilled water (dH2O). DMSO dissolved in dH2O was used as the vehicle.

To block c-Abl, mice were intraperitoneally treated with imatinib mesylate (5–150 mg kg−1 body weight day−1 dissolved in dH2O) (Wako) twice a day for 2 consecutive days before the administration of AK18, AK23, fluorescence-conjugated mouse IgG, or fluorescence-conjugated dextran. For the evaluation of hair loss, the development of hair loss in the back skin was evaluated for 10 days post AK23 administration, pretreated with imatinib or dH2O (n = 10 or 13, respectively). Alternatively, AICAR (Toronto Research Chemicals, Toronto, Canada) was dissolved in dH2O and subcutaneously administered 1 day at 0.5–0.75 mg kg−1 body weight day−1 before AK18 administration.

To block Pdgfr-α in vivo, mice were intravenously treated with 1 mg of APA5 (Cat. 562171, clone APA5, BD Bioscience) or 1 mg of control rat IgG 24 h before the administration of AK18.

To block FcRn function, IVIG at 1.0 g kg−1 day−1 (human IgG obtained from Nihon Pharmaceutical Co. Ltd, Tokyo, Japan) was injected intravenously via the tail vein for 3 consecutive days to adult mice or pregnant mice (from E15.5 to E17.5). AK18 was administered at 100 μg together with the final IVIG administration. For the analyses, the ears of adult mice were collected the following day. The fetuses were delivered at E18.5 by cesarean section, and the dorsal skin was extracted for cell isolation and flow cytometry.

**Cell culture and in vitro IgG endocytosis assay.** HDBECs were purchased from PromoCell (Heidelberg, Germany) and cultured in endothelial cell growth medium MV (PromoCell) at 37 °C in 5% CO2 until reaching 70–80% confluency. For the IgG endocytosis assay, cells were incubated in the presence of A594-conjugated human IgG (Cat. 009-580-003, 10 μg ml−1; Jackson ImmunoResearch) for 1 h. To block dynamin function, cells were treated with dynasore (100 μg ml−1; Santa Cruz Biotechnology, Dallas, TX) for 30 min. To block caveolae-mediated endocytosis, cells were cultured with nystatin (50 μg ml−1; Sigma Aldrich) for 30 min. To evaluate the quantity of engulfed IgG in each cell, the cytoplasmic area of cells in a high-power field (×40) was manually circumscribed using ImageJ[58], and the MFI of the signals from A594-IgG was measured. ΔA594-IgG was calculated by subtracting the background MFI.

**Immunocytochemistry of HDBECs.** For immunocytochemistry, HDBECs were incubated with A594-conjugated human IgG (Cat. 009-580-003, 10 μg ml−1; Jackson ImmunoResearch, West Grove, PA) for 1 h, fixed with 4% PFA for 1 h at 4 °C, permeabilized with 0.1% triton-X100 in 1% bovine serum albumin/PBS, and blocked with Image-iT FX signal enhancer (Invitrogen) before the staining. Subsequently, cells were incubated with anti-EEA-1 Ab (Cat.2411, Cell Signaling Technology, Danvers, MA) at room temperature for 1 h, washed and incubated with anti-rabbit IgG A488 dye (Invitrogen) for 30 min at room temperature, and mounted with ProLong Diamond Antifade Mountant with DAPI (Life Technologies). Alternatively, cells were incubated with HCS CellMask™ Green Stain (Cat. H32714, Invitrogen) at 2 μg ml−1 for 30 min at room temperature, and mounted. Otherwise, cells were stained with anti-caveolin 1 Ab (Cat. ab2910, Abcam) or anti-clathrin Ab (Cat. ab21679, Abcam) overnight at 4 °C, incubated with anti-rabbit IgG A488 dye (Invitrogen) for 30 min at room temperature on the following day, and mounted. For the acid wash[34], cells were washed with 0.2 M acetic acid in 150 mM NaCl at 4 °C after the IgG endocytosis assay and before the fixation. These slides were observed under a fluorescent microscope, BZ-900 (KEYENCE).

**Transwell permeability assay.** HDMECs (PromoCell) were purchased and cultured in endothelial cell growth medium MV (PromoCell) at 37 °C in 5% CO2 until reaching 70–80% confluency. HDMECs were collected with the DetachKit (Promocell) and plated on the insert plates of Millicell-96 transwell (Merck Millipore, Darmstadt, Germany) at 5 × 104 cells in 75 μl medium/well. Two hundred and fifty microliters of medium was added to the receiver plate and incubated for 2 days at 37 °C in 5% CO2. The insert plates were then replaced to the new receiver plate. The medium was aspirated from the insert plates, and new medium with imatinib,

asciminib, or GNF-2 was added at 50 μl/well at various concentrations as indicated in each figure, and incubated for 3 h. Asciminib and GNF-2 were synthesized in Mitsubishi Tanaka Pharma Corporation. Subsequently, 10 μl of FITC-conjugated human IgG at 187.5 μg ml$^{-1}$ (Jackson ImmunoResearch) was added to the insert plates, and incubated for 30 min. Ten microliters of thrombin was added at 5 U ml$^{-1}$ and incubated for another 60 min. Finally, the insert plates were removed, and fluorescence intensity of the FITC-conjugated human IgG in the receiver plates was measured by EnVision (PerkinElmer, Waltham, MA).

**In vitro siRNA assay**. The following control siRNA and siRNA that targets *caveolin 1* (*CAV1*), *clathrin heavy chain* (*CLTC*), or *c-ABL* (*ABL1*) were purchased from GE healthcare (Pollards Wood, UK): Accell human *CAV1* siRNA-SMART pool (Cat. E-003467-00-0005), Accell human *CLTC* siRNA-SMART pool (Cat. E-004001-00-0005), Accell human *ABL1* siRNA-SMART pool (Cat. E-003100-00-0005), and Accell Nontargeting Pool (Cat. D-001910-10-05). HDBECs (PromoCell) were treated with each siRNA following the manufacturer's protocol.

Briefly, HDBECs were cultured in endothelial cell growth medium MV (PromoCell) with 5% serum concentration, at 37 °C in 5% $CO_2$, until reaching 40–50% confluency. The medium was then exchanged to siRNA containing MV medium with 2.5% serum concentration, and cultured for another 3 days at 37 °C in 5% $CO_2$. Down regulation of each mRNA expression was verified at this time point by reverse transcription polymerase chain reaction. Subsequently, the medium was again exchanged to MV medium with 5% serum concentration, and cells were cultured for additional 3 days at 37 °C in 5% $CO_2$. After that, immunohistochemistry to check the down regulation of protein expression of caveolin 1 and clathrin, and the IgG endocytosis assay were performed as mentioned.

**Quantitative PCR analysis**. Total RNA was isolated from the cells lysed in buffer RLT by the RNeasy Mini kit (Qiagen, Hilden, Germany). cDNA was synthesized with a PrimeScript RT reagent kit and random hexamers according to the manufacturer's protocol (TaKaRa, Shiga, Japan). A LightCycler 480 and LightCycler SYBR Green I Master mix were used according to the manufacturer's protocol (Roche) for quantitative PCR (primer sequences are as follows: *CAV1*; Forward, CTGTCGGAGCGGGACATC/Reverse, TGTTTAGGGTCGCGGTTGAC. *CLTC*; Forward, GCACTGAAAGCTGGGAAAACT/Reverse, TCCGTAACAAGAGCA ACCGT. *ABL1*; Forward, GCTGTTATCTGGAAGAAGCCCT/Reverse, GCAACG AAAAGGTTGGGGTC). The expression of each gene was normalized to that of the internal control gene, *GAPDH* (Forward, CATGAGAAGTATGACAACAG CCT/Reverse, AGTCCTTCCACGATACCAAAGT).

**Electron microscopy**. Mice were anesthetized, and perfused with PBS containing heparin and then with half Karnovsky solution, containing 2% PFA and 2% glutaraldehyde (Wako) in 0.1 M cacodylic acid (Wako). Subsequently, the ears were cut into 1 × 3 mm pieces, prefixed with half Karnovsky solution overnight at 4 °C, and fixed in 1% osmium before the dehydration and epon embedding. Samples were then ultrathin-sliced and underwent uranium and lead staining, and observed using transmission electron microscope (H-7650, Hitachi High-Technologies Corporation, Tokyo, Japan).

**Statistics**. Unless otherwise indicated, data are presented as means ± SDs and each data point is representative of three independent experiments. As described in the "Methods" section, due to the limitation of the serum amount, the experiments using patients' sera were performed in a single mouse and its control, twice, with the similar results for each serum. Statistical analyses were performed using a GraphPad prism (GraphPad Software, Inc., CA). Normal distribution was assumed for all samples. Unless indicated otherwise, a parametric Student's *t*-test or a one-way ANOVA test was used for comparing groups and log-rank test was used for survival curves. A value of $P < 0.05$ at 95% confidence intervals was considered to indicate statistical significance. NS indicates not significant.

**Study approval**. All animal experimental procedures were ethically approved by the institutional Animal Care and Use Committee in Kyoto University Faculty of Medicine (182309). In all experiments, we complied with all relevant ethical regulations for animal testing and research. All experiments using human samples in this study was approved by the Medical Ethics Committee of the Kyoto University Graduate School of Medicine, and conducted according to the Declaration of Helsinki principles. Informed consent was obtained before the sampling of serum from each patient.

**Reporting summary**. Further information on research design is available in the Nature Research Reporting Summary linked to this article.

## Data availability
The source data underlying Figs. 1b, g, 2c, e–h, 3b–f, 4b–i, 5a–f, 5h, 6a–e, 7a, d, and Supplementary Figs. 3A, 4B, 4E, and 5B are provided as a Source Data file.

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

## Acknowledgements

This work was supported by the Japan Society for the Promotion of Science KAKENHI (201740146), Grants-in-Aid for Scientific Research 15H05790, 15H1155, 15K15417, Japan Science and Technology Agency, Precursory Research for Embryonic Science and Technology (PRESTO) (16021031300), and Japan Agency for Medical Research and Development (AMED) (16ek0410011h0003, 16he0902003h0002, 18ak0101057h0003). We thank Dr Y. Hamasaki (Kyoto University). We thank H. Doi, K. Tomari, N. Ishizawa, and M. Hiraiwa (Kyoto University), and K. Okamoto-Furuta and H. Kohda of Electron Microscopic Study, Center for Anatomical studies, Graduate School of Medicine (Kyoto University) for their technical assistance. We thank Z. Chow for language editing.

## Author contributions

Y.K. and T.T. produced transgenic mouse strains. S.O., G.E., T.H. and K.K. designed this study and wrote the paper; S.O., F.M., M.Y. and G.E. performed the experiments and analyzed data; M.A. developed experimental reagents; T.N., S.N., A.O., A.K., T.D., T.H. and K.K. directed the project and edited the paper; and all authors reviewed and discussed the paper.

## Additional information

**Competing interests:** The authors declare no competing interests.

