## [Peer Review File · Nature Communications]

Reviewers' comments:

Reviewer #1 (Remarks to the Author):

The major claims of the paper are that murine Pemphigus vulgaris (PV)-associated antibodies extravasate into the skin in homeostatic (non-inflammatory) conditions, mediated through dynamin-dependent endocytic vesicle formation, but not Fc-gamma receptors, in blood endothelial cells. Administration of the Abl family tyrosine kinase inhibitor imatinib impairs IgG extravasation in the skin through reduction of caveolae-mediated endocytosis and attenuates murine pemphigus manifestation. These findings are novel and will be of wider relevance to autoantibody-mediated autoimmune conditions in general. The authors took a careful step-wise approach to investigate mechanisms of IgG extravasation and the blocking thereof and the results provided are convincing. The results presented in this manuscript may encourage other studies and clinical trials aimed at inhibiting IgG extravasation. A potential shortcoming is that the choice of autoantibodies in the study is limited to those targeting murine Dsg3, while other targets in addition to Dsg3 have been described in PV; however, the pathways of antibody extravasation are likely independent of which autoAb is trafficked. Furthermore, it needs to be seen if targeting Abl family tyrosine kinases will be as effective in humans as in the murine model. Finally, it remains to be seen if the side effects observed after targeting Abl family tyrosine kinases (e.g. in cancer) outweigh the benefits in autoimmune diseases; particularly compared to recent advances in targeting the B-cell response (such as rituximab) in these conditions.

Generally, the descriptions of several experiments lack some detail that would enable other researchers to reproduce the work. Examples include but are not limited to:

- Results: Clarify the age of mice used and duration from Ak23 injection to hair loss. Clarify what is the relevance of measuring CD45-, E-cadherin+ keratinocytes.
- Results, Fig 1 E, F, G: Disclose time from injection to analysis.
- Results, Fig 2: Disclose timing of AK18 engulfment in dermal BECs after Ab injection.
- Results, Caveola mediated endocytosis is essential for IgG extravasation in the skin: Paragraph describes experiments after injection of fluorescein conjugated IgG. However, it switches to AK18 deposition in Fig. 3F. How was the latter experiment performed?

Minor comments:

- Introduction; Clarify that the murine model used in this study is a passive model that does not address initiation of the autoantibody response and target specificity.
- Introduction; PV is a prototypic antibody-mediated disorder that targets a keratinocyte adhesion molecule, Dsg3, among other targets (or has Dsg3 as its main target molecule).
- Methods, Cell isolation and Flow cytometry:"R&D systems (Toll Free USA, Canada)", should list city, country instead.
- throughout the text, use "sera" instead of "serums"

Reviewer #2 (Remarks to the Author):

In this report, Ono and colleagues explore the involvement of Abl tyrosine kinases on IgG extravasation.

There are some missing critical experiments and information, that preclude its publication at this stage.

- The authors need to clarify the number of animals used for each specific study (n animals, n group). Importantly, ethical statement on animal experimentation is missing.
- Data (Fig 5 and 6) were obtained only with drugs, again genetic proofs (siRNA, CRISPR) are required.

Other points are listed below:

Fig 1C. Why MFI mean is above 750 in left panel, while it is around at 400 in right panel?

Fig 1E. Staining in vehicle and Ig control animals need to be shown.

Fig 1F-G. Macrophage staining needs to be better explained.

Fig 2. What happen to PMA-induced leakage in D1D2 KO animals? What happen to the endothelium barrier and morphology in D1D2 KO animals, including EM analysis, matrix deposit and imaging of cell-cell junctions.

Fig 3A. Ig staining in A requires co-staining with internal markers (EEA1) and membrane, to make sure it is internalized. Acid wash will also demonstrate the internal localization.

Fig 3D. The authors need to show the effects of Nystatin on IgG internalization.

Fig 3E. The authors have to show caveolin siRNA efficiency.

Fig 3E-F. Ig internalization needs to be controlled in clathrin siRNA.

Reviewer #3 (Remarks to the Author):

The paper from Ono and colleagues bring a rather new important and neglected aspect of antibody biology to light. They show that antibody extravasation and tissue penetration, a very relevant process, both for antibody-pathologies (e.g. autoantibodies) and antibody therapies, is not dependent of FcRn as one would expect, activating or the inhibitory FcγR, but mysteriously through Abl-family of tyrosine kinases. Although the mechanistically this is still very puzzling, the current paper is still highly thought provoking and should stimulate others to stipulate exactly what is going on. There are many potential ways on how to improve the paper, but that may take years and will not advance this field fast enough, and may still not deliver the full answer. I have no major comments, except perhaps on the use of the English language, but I think this can be solved, but I have longer list of minor comments/corrections/suggestions and possible improvements.

Minor

-grammar and the use of English language needs to be improved.

-Page 4, lines 67-8: the notion that any cells probe Ig by Ig-receptors sound like a conscient act while it is a purely biochemical reaction – the affinity and on/off rates depends on the type of FcεR. Please amend to reflect this

-Line 88: please make a distinction here between the classical FcγR (mostly myeloid) and the Neonatal Fc receptor

-Page 5, lines 95-6: the sentence contains a complex set of double negatives. Please simplify, e.g. "...attributed to the decreased paracellular permeability by inhibiting the Abl family of tyrosine kinases."

-Page 6, line 132, please replace the word "murine" (meaning rodent) with the exact species.

-Line 132: please state clearly that you are referring to irrelevant (aspecific) IgG. For Figure 1 and in the main text it is not completely clear what Vehicle control consists of.

-Figure 2 legend. The use of the word entrapped is very subjective/suggestive. Please use another

more objective word

-In figure 3, it would have been even better to test directly if clathrin blocking also affects IgG extravasation/transport.

-Page 10 line 244: the activity is simply internalization of IgG, not phagocytic activity

-Line 270, "has been found to have a potency...", keep it past tense throughout. This is not the case in many places.

-A considerable discussion is intertwined in the result section, one notable section are lines 279-282, although I must admit the logic of the text will not be better if this text is moved to the discussion. However, I think you can simply remove this, and then continue in next paragraph "To further examine the possible effect of the Abl family of tyrosine..."

-Figure 6: many of the inhibitors have not been titrated to a level where a concentration dependent effect is seen. The effect of Asciminib, is much less than of the more specific inhibitors...

-Line 326: "...observation for AK18..."

-Page 13, results: -was there no pathological effect seen with the patient sera in the mice experiment (Fig S3)?? This would be nice to show (I agree the exp in figure 7 is very illustrative) or comment on that and why this is not the case

-In the experiments testing FcRn dependency, it would have been nicer to have the alpha-chain KO mice. However, as you show in the pregnancy model that this is very much FcRn/beta-2M dependent this is convincing. Alternatively, in some of your experiments it would also have been nice to add anti-FcRn blocking antibodies. Again, it is my opinion that the authors show clearly enough that FcRn is not involved.

-Discussion line 360: The line starting with "Thus, Fc portion of IgG..." is pure speculation. Please remove.

-Line 401: Probably the authors mean to say internalize circulating IgG??? Digest is the wrong word. Pinocytosis is known to be a normal process by which IgG enters these (and most) cells, also becoming accessible for FcRn.

Methods, page 468: it is not clear how IgG was measured on cells. Did you do an extracellular staining or after fix and perm??? You also mention trypsinization, does this not affect staining procedures?

Line -479: "anti"

Revise letter to the reviewers:

Thank you for revising our manuscript NCOMMS-19-06054, entitled “**Abl family tyrosine kinases govern homeostatic IgG extravasation in the skin in a murine pemphigus model**”.

First of all, we deeply appreciate the valuable comments of the reviewers on our manuscript. We took all the suggestion seriously and attempted to reply to all the comments. Our point-by-point responses to the comments are described on detailed response to reviewers, and modifications in the revised manuscript are highlighted by red color for the convenience of the reviewers.

Gyohei Egawa MD, PhD and Kenji Kabashima MD, PhD
Department of Dermatology
Kyoto University Graduate School of Medicine

Reviewers' comments:

Reviewer #1 (Remarks to the Author):

The major claims of the paper are that murine Pemphigus vulgaris (PV)-associated antibodies extravasate into the skin in homeostatic (non-inflammatory) conditions, mediated through dynamin-dependent endocytic vesicle formation, but not Fc-gamma receptors, in blood endothelial cells. Administration of the Abl family tyrosine kinase inhibitor imatinib impaires IgG extravasation in the skin through reduction of caveolae-mediated endocytosis and attenuates murine pemphigus manifestation. These findings are novel and will be of wider relevance to autoantibody-mediated autoimmune conditions in general. The authors took a careful step-wise approach to investigate mechanisms of IgG extravasation and the blocking thereof and the results provided are convincing. The results presented in this manuscript may encourage other studies and clinical trails aimed at inhibiting IgG extravasation. A potential shortcoming is that the choice of autoantibodies in the study is limited to those targeting murine Dsg3, while other targets in addition to Dsg3 have been described in PV; however, the pathways of antibody extravasation are likely independent of which autoAb is trafficked. Furthermore, it needs to be seen if targeting Abl family tyrosine kinases will be as effective in humans as in the murine model. Finally, it

remains to be seen if the side effects observed after targeting Abl family tyrosine kinases (e.g. in cancer) outweigh the benefits in autoimmune diseases; particularly compared to recent advances in targeting the B-cell response (such as rituximab) in these conditions.

We are greatly encouraged by the reviewer's comments and would like to improve our manuscript in his/her guidance.

Generally, the descriptions of several experiments lack some detail that would enable other researchers to reproduce the work. Examples include but are not limited to:

Results: Clarify the age of mice used and duration from AK23 injection to hair loss. Clarify what is the relevance of measuring CD45⁻, E-cadherin⁺ keratinocytes.

- Answer) The age of mice we used in AK23 trial was between 9-10 weeks old. Hair loss was started 4-5 days after AK23 injection, and progressed until 10 days after AK23 injection. The images of hair loss were obtained at day 10. We specified the information in the figure legends (Fig. 1A), the section of Method (Mice), and in the section of Method (Passive murine pemphigus model), respectively, as follows. We apologize for the lack of explanation.
-
- Page 17, line 469, Method (Mice): “The age of mice used in AK23 trials was between 9-10 weeks old.”
- Page 17, line 500, Method (Passive murine pemphigus model): “Hair loss started 4-5 days after AK23 injection, and progressed until 10 days after AK23 injection. The images of hair loss were obtained 10 days after AK23 injection”.
- Page 31, line 910, Figure 1: “A, Manifestation of hair loss in mice 10 days after intravenous injection of AK18 or AK23 at 10 µg or 100 µg.”
-
- For the second comment, the murine epidermis contains not only keratinocytes, but also Langerhans cells and T cells. We thus gated in CD45⁻ cells to exclude Langerhans cells and T cells. In addition, keratinocytes in both interfollicular epidermis (from basal layer to spinous layer) and follicular epidermis are reported to express E-cadherin (Christopher L, et al. *PNAS*. 2004;101 (2): 552-557, Young P, et al. *EMBO J*. 2003; 22(21): 5723–5733.), and the expression of E-cadherin was stable even after our flow cytometric procedure compared to another membranous marker of keratinocytes, such as CD49f. Therefore, we gated in E-cadherin⁺ cells to define keratinocytes in cell lysates (which may contain a lot of debris in CD45⁻ E-cadherin⁻ fraction). We added the information in the revised manuscript as follows:
-
- Page 6, line 122, Text: “Twenty-four hours later, the deposition of AK18 in the ear epidermis was quantified by measuring the mean fluorescence intensity (MFI) of IgG1 in the CD45⁻ E-cadherin⁺ keratinocytes by flow cytometry (hereafter referred to as IgG1-MFI) (Fig. 1B).”

Page 19, line 534, Method (Cell isolation and flow cytometry): “We gated CD45⁻ E-cadherin⁺ fraction to exclude intraepidermal immune cells and cell-debris.”

Results, Fig 1 E, F, G: Disclose time from injection to analysis.

Results, Fig 2: Disclose timing of AK18 engulfment in dermal BECs after Ab injection.

- Answer) Analyses were performed 24 h after AK18 injection in Fig. 1E, F, G, and Fig. 2B. In most experiments in the manuscript, analyses were performed 24 h after AK injections. This information can be found in the section of in the Method (the section of passive murine pemphigus model); “For a flow cytometric analysis, 100 µg of AK18 was intravenously administered to mice via the tail vein, and the ear skin was collected 24 h later unless otherwise indicated”, and we also added the information in each figure legend (Fig. 1E-G, and 2) in the revised manuscript.

Results, Caveola mediated endocytosis is essential for IgG extravasation in the skin:

Paragraph describes experiments after injection of fluorescein conjugated IgG. However, it switches to AK18 deposition in Fig. 3F. How was the latter experiment performed?

- Answer) The experiment of Fig. 3F was performed in the passive murine pemphigus model. We injected AK18 intravenously to mice 1 h after nystatin- or vehicle-pretreatment, and evaluated epidermal IgG1-MFI in the ears of mice. Due to the short half time of nystatin, we collected ear samples 6 h after AK18 administration in these trials. We added the sentence to explain the experiment in more detailed manner in the text and the Method (Reagents and *in-vivo* blocking assay) as follows:

- Page 8, line 208, Text: “To study the effect of nystatin *in vivo*, we injected nystatin intraperitoneally before the intravenous AK18 administration, and evaluated epidermal IgG1-MFI.”

- Page 21, line 604, Method (Reagents and *in-vivo* blocking assay): “To block caveolae-mediated endocytosis, mice were intraperitoneally treated with 8 mg kg⁻¹ body weight of nystatin (Sigma Aldrich) one hour before AK18 administration. Ear samples were evaluated for IgG1-MFI 6 h after AK18 administration. Nystatin was first dissolved in dimethyl sulfoxide (DMSO) at 25 mg ml⁻¹, and diluted to the final concentration of 1 mg ml⁻¹ in distilled water (dH₂O). DMSO dissolved in dH₂O was used as the vehicle.”

Minor comments:

Introduction; Clarify that the murine model used in this study is a passive model that does not address initiation of the autoantibody response and target specificity.

- Answer) We appreciate the suggestion, and added the following sentence in our revised manuscript:
- Page 5, line 104, Introduction: “With the passive murine model used in this study, we did not address initiation of autoantibody response or the target specificity, but simply focused on the kinetics of IgG extravasation *in vivo*.”

Introduction; PV is a prototypic antibody-mediated disorder that targets a keratinocyte adhesion molecule, Dsg3, among other targets (or has Dsg3 as it's main target molecule).

- Answer) We also appreciate this comment, and added the sentence in our manuscript as follows:

-

- Page 5, line 101: ” **Pemphigus vulgaris is a prototypic antibody-mediated disorder that targets a keratinocyte adhesion molecule, desmoglein (Dsg) 3, among other targets.**”

Methods, Cell isolation and Flow cytometry:”R&D systems (Toll Free USA, Canada)”, should list city, country instead.

- Answer) We corrected the description to “**R&D systems (Minneapolis, MN)**”.

-

throughout the text, use “sera” instead of “serums”

- Answer) We corrected all the term to “sera” in the revised manuscript.

Reviewer #2 (Remarks to the Author):

In this report, Ono and colleagues explore the involvement of Abl tyrosine kinases on IgG extravasation. There are some missing critical experiments and information, that preclude its publication at this stage.

The authors need to clarify the number of animals used for each specific study (n animals, n group). Importantly, ethical statement on animal experimentation is missing.

- Answer) We apologize for the missing information. We added the number of animals in each figure legend in the revised manuscript. The ethical statement on animal experiments can be found in the section of “Study approval” (We added the approval number for the additional information).

-

- Page 24, line 725, Study approval: “All animal experimental procedures were approved by the institutional Animal Care and Use Committee in Kyoto University Faculty of Medicine (182309). All experiments using human samples in this study was approved by the Medical Ethics Committee of the Kyoto University Graduate School of Medicine, and conducted according to the Declaration of Helsinki principles.”

-

Data (Fig 5 and 6) were obtained only with drugs, again genetic proofs (siRNA, CRISPR) are required.

- Answer) To respond this comment, we considered to perform *c-ABL* (*ABL1*)-siRNA to IgG transwell assay using HDMECs and IgG endocytosis assay using HDBECs.

First, unfortunately, we found that the treatment of *ABL1*-siRNA did not work on IgG transwell assay using HDMECs. After treatment with both non-targeting-siRNA group

and *ABL1*-siRNA group, the endothelial cell barrier on the transwell-membrane seemed to be disrupted, since the transition of IgG occurred even in the absence of thrombin stimulation. We thus considered that siRNA treatment is not suitable for *in-vitro* transwell assay because the treatment itself affects the barrier function of the membrane of endothelial cells.

We also considered to apply the CRISPER-Cas9 system to HDMECs. However, because HDMECs are primary human cells, which consist of heterogenous cell populations including lymphatic endothelial cells (LECs) and blood endothelial cells (BECs), the CRISPER-Cas9 system seemed to be unsuitable for gene editing of HDMECs. Thus, at this stage, we could not get genetic proofs of IgG transwell assay.

Next, we applied *c-ABL* (*ABL1*)-siRNA to IgG endocytosis assay using HDBECs. In IgG endocytosis assay, IgG endocytosis in *ABL1*-siRNA-treated group was attenuated compared to non-targeting siRNA control group, which was similar to the results of the experiment with *caveolin 1*-siRNA. We added this figure in Fig. 6A in the revised manuscript as follows:

Page 12, line 325, Text: “To further investigate the direct effect of Abl family tyrosine kinases on IgG extravasation, we examined IgG endocytosis by HDBECs that were exposed to siRNA against *c-ABL*. We confirmed that IgG endocytosis by HDBECs was decreased when cultured with siRNA against *c-ABL*, as well as siRNA against *caveolin 1* (Fig. 6A).”

Page 34, line 1010, Figure 6: “A, The left panels show hIgG distribution in BECs treated with siRNA that targets *caveolin 1* (*CAV1*), *c-ABL* (*ABL1*), or non-targeting siRNA. Scale bar = 20 μ m. The right panel shows Δ A594-MFI of BECs of each cell (n = 30, 22, and 20 cells, respectively), and BECs treated with siRNA that targets *caveolin 1* (*CAV1*), *c-ABL* (*ABL1*) or non-targeting siRNA (E, n = 22, 23, and 25 cells, respectively) after IgG endocytosis assay. *, P < 0.05.”

Figure 6A

-
- Prior to the experiment, the efficiency of *ABL1*-siRNA was verified in HDBECs by RT-PCR analysis. The result is presented as supplementary figure 3A in the revised manuscript as follows:
- Page 35, line 1059, Supplementary Figure 3: “**The levels of mRNA or protein expression after each siRNA treatment.**A, The levels of mRNA expression after siRNA treatment against *caveolin 1* (*CAVI*), *clathrin* (*CLTC*), *c-ABL* (*ABL1*), or non-targeting siRNA (n = 3 wells, respectively). **B**, The levels of protein expression evaluated by immunocytochemistry of HDBECs after each siRNA treatment. Green represents caveolin 1 or clathrin. Blue represents nuclei stained with DAPI. Scale bar = 20 μm .”

Supplementary Figure 3A

- We also improved the information of the siRNA assay and added the section of quantitative PCR analysis in the method section as follows:
-
- Page 23, line 671, Method (In-vitro siRNA assay): “**In-vitro siRNA assay.** The following control siRNA and siRNA that targets *caveolin 1* (*CAVI*), *clathrin heavy chain* (*CLTC*), or *c-ABL* (*ABL1*) were purchased from GE healthcare (Pollards Wood, UK): Accell human *CAVI* siRNA-SMART pool (Cat. E-003467-00-0005), Accell human *CLTC* siRNA-SMART pool (Cat. E-004001-00-0005), Accell human *ABL1* siRNA-SMART pool (Cat. E-003100-00-0005), and Accell Non-targeting Pool (Cat. D-001910-10-05). HDBECs (PromoCell) were treated with each siRNA following the manufacturer’s protocol.
- Briefly, HDBECs were cultured in endothelial cell growth medium MV (PromoCell) with 5% serum concentration, at 37°C in 5% CO₂, until reaching 40-50% confluency. The medium was then exchanged to siRNA containing MV medium with 2.5% serum

concentration, and cultured for another 3 days at 37°C in 5% CO₂. Down regulation of each mRNA expression was verified at this time point by reverse transcription polymerase chain reaction. Subsequently, the medium was again exchanged to MV medium with 5% serum concentration, and cells were cultured for additional 3 days at 37°C in 5% CO₂. After that, immunohistochemistry to check the down regulation of protein expression of caveolin 1 and clathrin, and IgG endocytosis assay were performed as mentioned.”

Page 23, line 690, Method (Quantitative PCR analysis): “**Quantitative PCR analysis**

- Total RNA was isolated from the cells lysed in buffer RLT by RNeasy Mini kit (Qiagen, Hilden, Germany). cDNA was synthesized with a PrimeScript RT reagent kit and random hexamers according to the manufacturer’s protocol (TaKaRa, Shiga, Japan). A LightCycler 480 and LightCycler SYBR Green I Master mix were used according to the manufacturer’s protocol (Roche) for quantitative PCR (primer sequences are as follows: *CAVI*; Forward, CTGTCGGAGCGGGACATC / Reverse, TGTTTAGGGTCGCGGTTGAC. *CLTC*; Forward, GCACTGAAAGCTGGGAAAACCT / Reverse, TCCGTAACAAGAGCAACCGT. *ABL1*; Forward, GCTGTTATCTGGAAGAAGCCCT / Reverse, GCAACGAAAAGGTTGGGGTC.). The expression of each gene was normalized to that of the internal control gene, *GAPDH* (Forward, CATGAGAAGTATGACAACAGCCT / Reverse, AGTCCTCCACGATACCAAAGT).”

Other points are listed below:

Fig 1C. Why MFI mean is above 750 in left panel, while it is around at 400 in right panel?

- Answer) We apologize for this confusion. The right panel next to Fig. 1C belonged to Fig. 1B in the original manuscript, but we realized that this layout was confusing. We rearranged the layout of Fig. 1B and 1C in the revised manuscript.

Fig 1E. Staining in vehicle and Ig control animals need to be shown.

- Answer) According to this comment, we added the staining of control animals (PBS i.v. and mouse IgG1 i.v.) to the revised figure (Fig. 1E) and the Method section (Immunohistochemistry of the murine ear skin) as follows:

Page 32, line 920, Figure 1: “E, Immunohistochemical evaluation in the mouse ear 24 h after intravenous injection of AK18 or AK23 at 10 µg or 100 µg, PBS, and 100 µg of control mouse IgG1 (Ctrl-mIgG1).”

Figure 1E

- Page 20, line 577, Method (Immunohistochemistry of the murine ear skin): “**PBS or control mouse IgG1 (Cat. 400166, clone MOPC-21, BioLegend) was injected as negative controls.**”

Fig 1F-G. Macrophage staining needs to be better explained.

- Answer) To respond this comment, we revised our manuscript in the text and the Method section as follows:
-
- Page 6, line 134, Text: “**In order to check whether non-specific IgG, not restricted to anti-Dsg3 Abs, extravasates from the blood to the dermal interstitium under homeostatic conditions, we intravenously injected fluorescein-conjugated non-specific mouse IgG to mice. Twenty-four hours later, by the whole mount immunohistochemistry of the split ear skin, we found that dermal macrophages were labeled, suggesting that they received extravasated IgG in the dermis (Fig. 1F and 1G).**”
-
- Page 20, line 591, Method (Immunohistochemistry of the murine ear skin); “To detect the extravasation of non-specific IgG or dextran in the murine ears by immunohistochemistry, 100-200 µg of fluorescein (A555 or A594)-conjugated mouse IgG (Cat. 015-160-003 or 015-600-003, Jackson ImmunoResearch, West Grove, PA) and/or 1 mg of FITC-conjugated 150 kDa dextran (Sigma Aldrich) was intravenously injected via the tail vein to mice. Otherwise, 20 µg of fluorescein-conjugated mouse IgG was intradermally injected to the ears. **The ears of mice were collected 24 h after injection and the dorsal halves of the ears were floated on 4% PFA at 4°C for a day. The ears were mounted with ProLong Diamond Antifade Mountant without DAPI (Life Technologies, Carlsbad, CA).**”
- Page 19, line 541, Method (Cell isolation and flow cytometry): “To detect the extravasation of non-specific IgG or dextran in the murine ears by flow cytometry, 100-200 µg of fluorescein (A647)-conjugated **mouse** IgG (Cat. 015-600-003, Jackson ImmunoResearch, West Grove, PA) and/or 1 mg of FITC-conjugated 150 kDa dextran (Sigma Aldrich) was intravenously injected via tail vein to mice or was intradermally injected to the ears of mice. **The dermal cell suspension was collected after extraction of epidermis with dispase II, and subsequently digested with liberase TL, as mentioned above. Cells were stained with fixable viability dye, anti-mouse CD45, CD11b, MHC classII, and F4/80. Cells were then fixed with Cytotfix/Cytoperm for flow cytometric analysis.**”

Fig 2. What happen to PMA-induced leakage in D1D2 KO animals? What happen to the endothelium barrier and morphology in D1D2 KO animals, including EM analysis, matrix deposit and imaging of cell-cell junctions.

- Answer)

To answer the questions, we performed several additional experiments.

First, PMA-induced leakage in D1D2 KO mice was evaluated by two-photon microscopy. The leakage was obviously unchanged in Cre⁺ mice compared to that of Cre⁻ control mice. We also evaluated the ear swelling and IgG1-MFI of the ear epidermis and found that they were comparable between Cre⁺ and Cre⁻ mice. We added these results as supplementary Fig. 1B, Fig. 2G, and 2H, in the revised manuscript, respectively.

Second, we evaluated the morphology of BECs by electron microscopy. Intriguingly, we found that the vesicles in dermal BECs were almost diminished in Cre⁺ D1D2 KO mice in both homeostatic and inflammatory conditions. On the other hand, the interendothelial junction between endothelial cells was maintained in Cre⁺ mice in homeostatic condition as well as that of control Cre⁻ mice. Even in the inflammatory condition, no obvious interendothelial gaps were observed in both Cre⁺ and Cre⁻ mice, but we could not exclude the possibility that this observation was due to the limitation of number of vessels that we could observed. As for the matrix deposits, we found the reduction of IgG positive macrophages in Cre⁺ D1D2 KO mice compared to Cre⁻ controls after intravenous injection of fluorescence-conjugated IgG in homeostatic condition. These results are presented as Fig. 2D and Supplementary Fig. 1A in the revised manuscript.

Taken together, we concluded that the paracellular leakage induced by topical PMA application is unaffected in VEcadherin-CreERT2; D1D2-floxed mice. In addition, the results of electron microscopy and matrix deposit assay supported our hypothesis that transcellular IgG transport is crucial for IgG extravasation in homeostatic condition. Furthermore, we considered that the disability of transcellular transportation in VEcadherin-CreERT2; D1D2-floxed mice is negligible in total IgG extravasation when the robust paracellular leakage occurs upon inflammation.

Given that imatinib reduced IgG1-MFI even in the inflamed ears, we considered that imatinib may also affected the paracellular leakage of IgG. The ear swelling after topical PMA application was, however, unchanged both in imatinib-treated mice and VEcadherin-CreERT2; D1D2-floxed mice compared to those of their controls. We considered that this might be due to the difficulty in evaluating the extent of paracellular permeability by the ear swelling measurement.

We revised the manuscript to include above discussions and added Fig. 2D, 2G, 2H and Supplementary Fig. 1, as follows:

-
- Page 7, line 167, Text: **“We confirmed by electron microscopy that vesicles in dermal BECs were almost diminished in Cre⁺ mice after tamoxifen treatment compared to Cre⁻ control mice (Fig. 2D, red arrowheads). The interendothelial adhesion junction**

was unaffected in Cre⁺ mice as well as Cre⁻ control mice under homeostatic condition (**Fig. 2D, yellow arrows**). In such condition, both epidermal AK18 deposition (evaluated by IgG1-MFI) and internalization of AK18 in dermal BECs were reduced (**Fig. 2E and 2F**). Consistently, the enhancement of dermal macrophages after intravenous injection of fluorescein-conjugated IgG was also attenuated in Cre⁺ mice, and IgG retained in the blood circulation (**Supplementary Fig. 1A**).”

Page 8, line 178, Text: “On the other hand, upon inflammation with topical PMA application to the ears, the ear swelling levels was unaffected (**Fig. 2G**), and the leakages of IgG to the interstitium were comparable between Cre⁺ and Cre⁻ control mice (**Supplementary Fig. 1B**), after the intravenous injection of fluorescein-conjugated IgG. In addition, epidermal AK18 deposition of Cre⁺ mice was also comparable to that of control mice in inflamed ears (**Fig. 2H**). These results suggest that the robust increase in the paracellular permeability upon inflammation is unaffected in VE-cadherin-CreERT2; D1D2-floxed mice, and that the defect of transcellular IgG extravasation is negligible in such situations. Together with our previous report which exhibiting that the epidermal IgG deposition is markedly high in inflamed-ears as compared to untreated-ears⁶, we considered that the transcellular pathway is less important for IgG extravasation in the presence of robust paracellular leakage under inflammatory conditions.”

Page 10, line 266, Text: “Therefore, we assumed that imatinib regulated, at least in part, the paracellular permeability of IgG in our model. This observation is consistent with previous reports exhibiting the impact of imatinib on vascular permeability under inflammatory conditions. The PMA-induced ear swellings were unaffected by imatinib (**Fig. 5B**), probably due to the fact that the amount of ear swelling is determined not only by vascular permeability..”

- Page 32, line 938, Figure 2: “**D**, Electron microscopic images of dermal BECs in the ear skin evaluated in VE-cadherin-CreERT2; D1D2-floxed mice in homeostatic condition. Red arrowheads show intracellular vesicles. Yellow arrows show interendothelial junctions. Scale bar = 500 nm. **E-F**, Epidermal IgG1-MFI (**E**) and the % frequency of IgG1-positive BECs (**F**) 24 h after 100 µg of intravenous AK18 injection or PBS in VE-cadherin-CreERT2; D1D2-floxed mice (n = 5 for Cre⁻ or n = 4 for Cre⁺, both ears were evaluated separately in **E**). *, *P* < 0.05. **G-H**, The ear thickness changes (**G**) and epidermal IgG1-MFI (**H**), 24 h after 20 µg of intravenous AK18 injection or PBS and topical PMA application to the ears, in VE-cadherin-CreERT2; D1D2-floxed mice (n = 6 for Cre⁻ and n = 5 for Cre⁺).
- **Figure 2**

- Page 36, line 1042, Supplementary Figure 1: “**IgG extravasation in the skin of VE-cadherin-CreERT2; D1D2-floxed mice. A,** Immunohistochemical images of ear skin dermis 24 h after intravenous injection of A594-conjugated mouse IgG. Scale bar = 100 µm. **B,** Time-lapse images of paracellular IgG leakage to the interstitium of the ear dermis of VE-cadherin-CreERT2; D1D2-floxed mice (15 min after intravenous injection of fluorescein-conjugated IgG via the tail vein). Scale bar = 150 µm.”

Supplementary Figure 1

The method for electron microscopy was also added in the revised manuscript as follows:

- Page 24, line 705, Method (Electron microscopy): “**Electron microscopy** Mice were anesthetized, and perfused with PBS containing heparin and then with half Karnovsky solution, containing 2% PFA and 2% glutaraldehyde (Wako) in 0.1 M cacodylic acid (Wako). Subsequently, the ears were cut into 1 x 3 mm pieces, prefixed with half Karnovsky solution overnight at 4°C, and fixed in 1% osmium before the dehydration and epon embedding. Samples were then ultrathin-sliced and underwent uranium and lead staining, and observed using transmission electron microscope (H-7650, Hitachi High-Technologies Corporation, Tokyo, Japan).”

Fig 3A. Ig staining in A requires co-staining with internal markers (EEA1) and membrane, to make sure it is internalized. Acid wash will also demonstrate the internal localization.

- Answer) As the reviewer suggested, we first performed co-staining with anti-EEA1 Ab (Cat.2411, Cell Signaling Technology) of HDBECs after IgG endocytosis assay.

We found that not all, but most IgG dots co-localized with EEA marker. We also tried staining of plasma membrane and cytoplasm using CellMask Green (Cat. H32714, Invitrogen), and found that all IgG-dots are actually included inside the cytoplasm. We also treated cells with acid wash (Koivusalo M et al, *J Cell Biol.* 2010: 188; 547–563) and found that IgG-dots in HDBECs was not obviously lost after the acid wash. Taken together, we concluded that most IgG-dots exhibited in our figures represent internalized-IgG. We added the information as Fig. 3A and Supplementary figure 2.

- Page 32, line 950, Figure 3: “**A**, Subcellular human IgG (hIgG) distribution in BECs (red) **co-stained with anti-EEA1 Ab (green)**. Blue represents a nucleus stained with DAPI. Scale bar = 10 μm ”

Figure 3A

- Page 35, line 1052, Supplementary Figure 2: “**Internalization of IgG in HDBECs.**
- **A**, Subcellular human IgG (hIgG) distribution in HDBECs (red), co-stained with anti-EEA1 Ab (green) (the upper panels). Blue represents a nucleus stained with DAPI. HDBECs stained without anti-EEA1 first Ab (the middle panels), and HDBECs without IgG endocytosis assay (the lower panels) were evaluated as negative controls for each staining. Scale bar = 10 μm . **B**, HDBECs stained with CellMask Green Stain after IgG endocytosis assay. Scale bar = 20 μm . **C**, HDBECs with or without acid wash after IgG endocytosis assay. Scale bar = 20 μm .”

Supplementary Figure 2

Fig 3D. The authors need to show the effects of Nystatin on IgG internalization.

- Answer) We added these data in Fig 3D in the revised manuscript.

Fig 3E. The authors have to show caveolin siRNA efficiency.

Fig 3E-F. Ig internalization needs to be controlled in clathrin siRNA.

- Answer) We appreciate the comments. The efficiency of each siRNA was verified by down regulation of mRNA with RT-PCR analysis and protein expression with immunohistochemistry, respectively as shown in Supplementary Fig. 3 in the revised manuscript. The precise methods for siRNA assay and RT-PCR reaction were added in the Method section.

- As suggested, we applied *clathrin* (*CLTC*)-siRNA in IgG endocytosis assay. IgG endocytosis in *CLTC*-siRNA-treated group was comparable to that of non-targeting siRNA-treated control group, while it was attenuated in *caveolin 1* (*CAVI*)-siRNA-treated group. The result is added in the text and as Fig. 3E as follows:

- Page 8, line 205, Text: “In addition, IgG endocytosis was suppressed in HDBECs with a caveolae-conformation inhibitor, nystatin³⁵ (**Fig. 3D**), or siRNA against *caveolin 1* but not siRNA against *clathrin* (**Fig. 3E and Supplementary Fig. 3**).”

- Page 32, line 957, Figure 3: “**D-E**, The left panels show hIgG distribution in nystatin- or vehicle-treated BECs (**D**), and BECs treated with siRNA that targets *caveolin 1* (*CAVI*), *clathrin* (*CLTC*) or non-targeting siRNA (**E**). Scale bar = 20 μ m. The right panel shows Δ A594-MFI of BECs treated with nystatin or vehicle (DMSO) (**D**, n = 9 and 10 cells, each), and BECs treated with siRNA that targets *caveolin 1* (*CAVI*), *clathrin* (*CLTC*) or non-targeting siRNA (**E**, n = 22, 23 and 25 cells, respectively) after IgG endocytosis assay. *, $P < 0.05$.”

- **Figure 3E**

- Page 35, line 1059, Supplementary Figure 3: **“The levels of mRNA or protein expression after each siRNA treatment. A, The levels of mRNA expression after siRNA treatment against *caveolin 1* (*CAV1*), *clathrin* (*CLTC*), *c-ABL* (*ABL1*), or non-targeting siRNA (n = 3 wells, respectively). B, The levels of protein expression evaluated by immunocytochemistry of HDBECs after each siRNA treatment. Green represents caveolin 1 or clathrin. Blue represents nuclei stained with DAPI. Scale bar = 20 μm .”**
-
- **Supplementary Figure 3**

Supplemental Figure 3

- Page 23, line 671, Method (In-vitro siRNA assay): “The following control siRNA and siRNA that targets *caveolin 1* (*CAVI*), *clathrin heavy chain* (*CLTC*), or *c-ABL* (*ABL1*) were purchased from GE healthcare (Pollards Wood, UK): Accell human *CAVI* siRNA-SMART pool (Cat. E-003467-00-0005), Accell human *CLTC* siRNA-SMART pool (Cat. E-004001-00-0005), Accell human *ABL1* siRNA-SMART pool (Cat. E-003100-00-0005), and Accell Non-targeting Pool (Cat. D-001910-10-05). HDBECs (PromoCell) were treated with each siRNA following the manufacturer’s protocol.
- Briefly, HDBECs were cultured in endothelial cell growth medium MV (PromoCell) with 5% serum concentration, at 37°C in 5% CO₂, until reaching 40-50% confluency. The medium was then exchanged to siRNA containing MV medium with 2.5% serum concentration, and cultured for another 3 days at 37°C in 5% CO₂. Down regulation of each mRNA expression was verified at this time point by reverse transcription polymerase chain reaction. Subsequently, the medium was again exchanged to MV medium with 5% serum concentration, and cells were cultured for additional 3 days at 37°C in 5% CO₂. After that, immunohistochemistry to check the down regulation of protein expression of caveolin 1 and clathrin, and IgG endocytosis assay were performed as mentioned. ”
-
- Page 23, line 690, Method (Quantitative PCR analysis): “Total RNA was isolated from the cells lysed in buffer RLT by RNeasy Mini kit (Qiagen, Hilden, Germany). cDNA was synthesized with a PrimeScript RT reagent kit and random hexamers according to the manufacturer’s protocol (TaKaRa, Shiga, Japan). A LightCycler 480 and LightCycler SYBR Green I Master mix were used according to the manufacturer’s protocol (Roche) for quantitative PCR (primer sequences are as follows: *CAVI*;

Forward, CTGTCGGAGCGGGACATC / Reverse, TGTTTAGGGTCGCGGTTGAC.
CLTC; Forward, GCACTGAAAGCTGGGAAAACCT / Reverse,
TCCGTAAACAAGAGCAACCGT. *ABL1*; Forward,
GCTGTTATCTGGAAGAAGCCCT / Reverse, GCAACGAAAAGGTTGGGGTC.).
The expression of each gene was normalized to that of the internal control gene,
GAPDH (Forward, CATGAGAAGTATGACAACAGCCT / Reverse,
AGTCCTTCCACGATACCAAAGT).”

Reviewer #3 (Remarks to the Author):

The paper from Ono and colleagues brings a rather new important and neglected aspect of antibody biology to light. They show that antibody extravasation and tissue penetration, a very relevant process, both for antibody-pathologies (e.g. autoantibodies) and antibody therapies, is not dependent of FcRn as one would expect, activating or the inhibitory FcγR, but mysteriously through Abl-family of tyrosine kinases. Although the mechanistically this is still very puzzling, the current paper is still highly thought provoking and should stimulate others to stipulate exactly what is going on. There are many potential ways on how to improve the paper, but that may take years and will not advance this field fast enough, and may still not deliver the full answer. I have no major comments, except perhaps on the use of the English language, but I think this can be solved, but I have longer list of minor comments/corrections/suggestions and possible improvements.

- Answer) We greatly appreciate all the comments, corrections, and suggestions. We are encouraged by the reviewer's comments and would like to improve our manuscript in his/her guidance.

Minor

-grammar and the use of English language needs to be improved.

- Answer) As suggested, our manuscript was checked by a native English speaker and some grammatical errors were corrected.

-Page 4, lines 67-8: the notion that any cells probe Ig by Ig-receptors sound like a conscient act while it is a purely biochemical reaction – the affinity and on/off rates depends on the type of FcεR. Please amend to reflect this.

- Answer) To reflect the suggestion, we avoid the use of “probe” and revised our manuscript as follows.
- Page 4, line 67, Text: “It was reported that blood-circulating IgE can bind to the high-affinity Fcε receptors of dermal perivascular mast cells, which elongate their dendrites through the blood vessel wall ⁵.”

-Line 88: please make a distinction here between the classical FcγR (mostly myeloid) and the Neonatal Fc receptor

- Answer) We revised the sentence as follows in the revised manuscript.
-
- Page 4, line 89, Text: “Although these studies implicate the importance of **FcRn**, **Fcγ** receptors, and endocytic vesicles for subcellular trafficking of IgG, their actual contribution to IgG extravasation in vivo has not been addressed.”
-
- Page 5, lines 95-6: the sentence contains a complex set of double negatives. Please simplify, e.g. “...attributed to the decreased paracellular permeability by inhibiting the Abl family of tyrosine kinases.”
- Answer) We appreciate the comment and revised our manuscript as suggested.
- Page 6, line 132, please replace the word “murine” (meaning rodent) with the exact species.
- Answer) We replaced the word “murine” to “mouse” throughout the text, as we have used mouse IgG reagents.
-
- Line 132: please state clearly that you are referring to irrelevant (aspecific) IgG. For Figure 1 and in the main text it is not completely clear what Vehicle control consists of.
- Answer) We added the sentence to clearly refer to the use of non-specific IgG in the experiments. We revised our manuscript as follows:
-
- Page 6, line 134: “**In order to check whether non-specific IgG, not restricted to anti-Dsg3 Abs, extravasates from the blood to the dermal interstitium under homeostatic conditions, we intravenously injected fluorescein-conjugated non-specific mouse IgG to mice. Twenty-four hours later, by the whole mount immunohistochemistry of the split ear skin, we found that dermal macrophages were labeled, suggesting that they received extravasated IgG in the dermis (Fig. 1F and 1G).**”
-
- In addition, we replaced the term “vehicle” to the exact solvents we used in the figure legend of each experiment (such as PBS, dH₂O, and DMSO). Especially, for *in-vivo* nystatin experiment (Fig. 3F), we described the content of vehicle in the Method section as follows:
-
- Page 21, line 604, Method (Reagents and in-vivo blocking assay): “To block caveolae-mediated endocytosis, mice were intraperitoneally treated with 8 mg kg⁻¹ body weight of nystatin (Sigma Aldrich) one hour before AK18 administration. **Ear samples were evaluated for IgG1-MFI 6 h after AK18 administration. Nystatin was first dissolved in dimethyl sulfoxide (DMSO) at 25 mg ml⁻¹, and diluted to the final concentration of 1 mg ml⁻¹ in distilled water (dH₂O). DMSO dissolved in dH₂O was used as the vehicle.**”
-
- Figure 2 legend. The use of the word entrapped is very subjective/suggestive. Please use another more objective word

- Answer) We replaced the word “entrapped” to “positive” throughout the text and revised our sentences as follows:
- “The % frequency of IgG1-positive BECs 24 h after intravenous AK18 injection or PBS, compared to isotype-stained control.”

-In figure 3, it would have been even better to test directly if clathrin blocking also affects IgG extravasation/transport.

- Answer) We appreciate the comment. We first intended to block clathrin function *in vivo*, but could not find any specific reagent. Thus, we additionally applied *clathrin (CLTC)*-siRNA to IgG endocytosis assay. IgG endocytosis in *CLTC*-siRNA-treated group was equivalent to that of non-targeting siRNA-treated control group, while it was attenuated in *caveolin 1 (CAVI)*-siRNA-treated group. The efficiency of both siRNA was also verified by down regulation of mRNA with RT-PCR analysis or protein expression with immunohistochemistry, respectively. The result is added as Fig. 3E and Supplementary Fig. 3 as follows:

- Page 32, line 957, Figure 3: “**D-E**, The left panels show hIgG distribution in nystatin- or vehicle-treated BECs (**D**), and BECs treated with siRNA that targets *caveolin 1 (CAVI)*, *clathrin (CLTC)* or non-targeting siRNA (**E**). Scale bar = 20 μm. The right panel shows ΔA594-MFI of BECs treated with nystatin or vehicle (DMSO) (**D**, n = 9 and 10 cells, each), and BECs treated with siRNA that targets *caveolin 1 (CAVI)*, *clathrin (CLTC)* or non-targeting siRNA (**E**, n = 22, 23 and 25 cells, respectively) after IgG endocytosis assay. *, $P < 0.05$.”

- **Figure 3E**

-

- Page 35, line 1059, Supplementary Figure 3: “The levels of mRNA or protein expression after each siRNA treatment. **A**, The levels of mRNA expression after siRNA treatment against *caveolin 1 (CAVI)*, *clathrin (CLTC)*, *c-ABL (ABL1)*, or non-targeting siRNA (n = 3 wells, respectively). **B**, The levels of protein expression

evaluated by immunocytochemistry of HDBECs after each siRNA treatment. Green represents caveolin 1 or clathrin. Blue represents nuclei stained with DAPI. Scale bar = 20 μm .”

-

- **Supplementary Figure 3**

-

-Page 10 line 244: the activity is simply internalization of IgG, not phagocytic activity

- Answer) To follow the suggestion, we replaced the word “engulfment by dermal BECs” to “**internalization of AK18 in dermal BECs**” throughout the manuscript.

-Line 270, “has been found to have a potency...”, keep it past tense throughout. This is not the case in many places.

- Answer) We appreciate the correction. We revised our manuscript to keep past tense throughout.

-

-A considerable discussion is intertwined in the result section, one notable section are lines 279-282, although I must admit the logic of the text will not be better if this text is moved to the discussion. However, I think you can simply remove this, and then continue in next paragraph “To further examine the possible effect of the Abl family of tyrosine...”

- Answer) We appreciate this suggestion and removed the line 279-282 from the text. We started the next paragraph with “**To further investigate the direct effect of Abl family tyrosine kinases on,,,**”

-Figure 6: many of the inhibitors have not been titrated to a level where a concentration dependent effect is seen. The effect of Asciminib, is much less than of the more specific inhibitors...

- Answer) As the reviewer pointed out, asciminib is the more specific inhibitor of c-Abl tyrosine kinase. From our examination, it seemed that imatinib and GNF-2 have stronger effect to inhibit IgG transportation than asciminib *in vitro*. From these results, we considered that Arg, another family of Abl kinase, might additionally play a role in inhibiting IgG-transportation through BECs. We also considered that there might be other unknown targets than c-Abl and Arg, on which imatinib have the effect. Since imatinib showed a strongest effect in mouse *in vivo*, we are now searching for the unknown target molecules of imatinib in BECs in cooperation with Mitsubishi Tanabe Pharma Corporation, as a research part of clinical trial of imatinib in patients with autoimmune-blistering diseases in Kyoto University Hospital (The study is registered with the UMIN Clinical Trials Registry, number UMIN000030865). We hope to find new targets from such further trials.

-Line 326: "...observation for AK18..."

Answer) We corrected "in" to "for".

-Page 13, results: -was there no pathological effect seen with the patient sera in the mice experiment (Fig S3)?? This would be nice to show (I agree the exp in figure 7 is very illustrative) or comment on that and why this is not the case

- Answer) Since we focused on Ab deposition in the experiment using the patient sera, we sacrificed mice 24 h after the administration of patients' sera. Hair loss might be prevented/occur in these mice with/without imatinib several days later. Unfortunately, the amount of patient sera was limited and we could not make evaluation for hair loss in those trials. We believe our clinical trial in patients with bullous pemphigoid as mentioned above will give favorable results.

-In the experiments testing FcRn dependency, it would have been nicer to have the alpha-chain KO mice. However, as you show in the pregnancy model that this is very much FcRn/beta-2M dependent this is convincing. Alternatively, in some of your experiments it would also have been nice to add anti-FcRn blocking antibodies. Again, it is my opinion that the authors show clearly enough that FcRn is not involved.

- Answer) We appreciate this suggestion. To respond this, we looked for commercially available anti-FcRn blocking antibody for mice, but we could not find it. Therefore, we applied intravenous immunoglobulin (IVIG) treatment as an alternative way to block FcRn function *in vivo*. As we have already reported (Ono et al. J. Allergy Clin. Immunol. 2018 141, 2273–2276.e1), IVIG at the dosage of 1.0 g/kg/day could completely prevent transplacental AK18 transfer to the fetuses. In this condition, however, AK18 extravasation and deposition to the ear skin of the adult mice was not prevented, which was comparable to that of control mice at 6 h after AK18 i.v. and partially decreased at 24 h after AK18 i.v. These results are quite similar to those of B2m-deficient mice. We consider that these data support our idea that FcRn is unnecessary for the transvascular IgG transfer in the skin. We added this information as Fig. 4G-I in the revised manuscript as follows:

- Page 10, line 243, Text: “In addition, we applied intravenous immunoglobulin (IVIG) treatment as an alternative way to block FcRn function. As we previously reported³⁹, IVIG at the dosage of 1.0 g kg⁻¹ day⁻¹ could completely prevent the transplacental AK18 transfer to fetuses (**Fig. 4G**). In this condition, however, epidermal AK18 deposition in the adult mice was not prevented: epidermal AK18 deposition and the serum AK18 levels were comparable to that of control mice at 6 h after AK18 administration (**Fig. 4H**), and both were partially decreased at 24 h after AK18 administration (**Fig. 4I**). These results were compatible with the results of β 2m-deficient mice (**Fig. 4D-F**), and also supported our idea that FcRn is unessential for transvascular IgG transport in the skin.”
- Page 33, line 981, Figure 4: “**G**, IgG1-MFI of fetuses from pregnant mice after intravenous AK18 injection, pretreated with 1.0 g kg⁻¹ day⁻¹ of IVIG or PBS (n = 7 or 6, each). PBS-injected mothers were used as the control (n = 4). **H**, IgG1-MFI of ear epidermis (the left panel, n = 4, each. Both ears were evaluated separately) or serum AK18 levels (the right panel, n = 3) in adult mice 6 h after intravenous AK18 injection, treated with IVIG at 1.0 g kg⁻¹ day⁻¹ or PBS. **I**, IgG1-MFI of ear epidermis (the left panel, n = 4, each) or serum AK18 levels (the right panel, n = 3, each) in adult mice 24 h after intravenous AK18 injection, treated with IVIG at 1.0 g kg⁻¹ day⁻¹ or PBS.”

- **Figure 4G-I**

-Discussion line 360: The line starting with “Thus, Fc portion of IgG...” is pure speculation. Please remove.

- Answer) We removed the sentence as suggested.

-Line 401: Probably the authors mean to say internalize circulating IgG??? Digest is the wrong word. Pinocytosis is known to be a normal process by which IgG enters these (and most) cells, also becoming accessible for FcRn.

- Answer) We considered that after the pinocytosis by BECs, some portion of IgG is transcytosed, some is recycled back to blood-circulation, and some is degraded in the lysosomal pathway. Because the pinocytosis step might be abrogated by imatinib, we considered both the recycling pathway and the degradation pathway in BECs stopped, resulting in the comparable serum AK18 titers in imatinib- and vehicle-treated group. We revised the revised manuscript as follows:

-
- Page 16, line 446, Text: “Although the definite mechanism is still undiscovered, it seems to be consistent with the notion that BECs not only recycle but also **degrade the internalized IgG**^{23,53}.”
-

Methods, page 468: it is not clear how IgG was measured on cells. Did you do an extracellular staining or after fix and perm??? You also mention trypsinization, does this not affect staining procedures?

- Answer) For IgG1 detection on the membrane of keratinocytes (epidermal IgG1-MFI), we performed membranous staining before the cell fixation. For IgG1 detection in blood endothelial cells (BECs), we performed intracellular staining of cells with IgG1 after fixation and permeabilization. During cell incubation with digestion enzymes, we added brefeldin A not to lose internalized-IgG in BECs. As the reviewer commented, long time trypsinization of the samples dampened the signal for IgG1 detection, thus we strictly limited the time for trypsinization to 8 min (The information was already described in the previous version of Method). The detailed information for digestion is added in the Method section (Cell isolation and flow cytometry) as follows:
-

- Page 18, line 515, Method (Cell isolation and flow cytometry): “**Brefeldin A (final concentration =10 ng ml⁻¹; Sigma Aldrich) was added to dispase II and liberase TL solution in order to evaluate the amount of intercellular AK18 in BECs.**”
-

- Page 18, line 532, Method (Cell isolation and flow cytometry): “**To check IgG1-MFI in the epidermis, epidermal cells were stained with antibodies against CD45, E-cadherin, and IgG1, then fixed with Cytfix/Cytoperm (Cat. 51-2090KZ, BD Bioscience), and underwent flow cytometric analysis. We gated CD45⁻ E-cadherin⁺ fraction to exclude intraepidermal immune cells and cell-debris. To evaluate cell-internalized IgG1 in dermal BECs, dermal cells were stained with fixable viability dye, antibodies against CD31, gp38, Ter119, and CD45. Cells were then fixed and permeabilized with Cytfix/Cytoperm and Perm wash (Cat. 51-2091KZ, BD Bioscience), and underwent intracellular-staining with anti-mouse IgG1 antibody for flow cytometric analysis.**”

Line -479: “anti”

- Answer) We corrected the sentence as suggested.

REVIEWERS' COMMENTS:

Reviewer #1 (Remarks to the Author):

The authors have responded to all concerns raised by this and the other reviewers. The manuscript is now suitable for publication.

Reviewer #2 (Remarks to the Author):

The authors did significantly improve the manuscript, I do not have further comments.

Reviewer #3 (Remarks to the Author):

The authors have responded adequately to all my comments and suggestions. However, I still notice that the level of English is still not completely suitable yet for publication. This applies especially to the new text, but also, as I noticed before, to the original text (which does not seem to have much improved, if at all).

Please consider the following changes:

Line 167 remove the word "almost"

Lines 186-7: "...previous report showing that epidermal IgG deposition was markedly increased in inflamed-ears compared to untreated-ears⁶, we concluded that the transcellular IgG-extravasation pathway is less important under inflammatory conditions when paracellular leakage is dominant."

Lines 195-198: Consider "IgG in HDBECs colocalized with EEA1, a marker of early endosomes, in foci within the cytoplasm (stained with CellMask Green), while being resistant to the acid wash by acetic acid³⁴, suggesting the IgG was internalized in HDBECs (Fig. 3A and Supplementary Fig. 2)."

Responses to the reviewers' comment

We really thank them kindly handling our manuscript. We appreciate all the reviewer's comments, and revised our manuscript to follow the request of reviewer 3.

REVIEWERS' COMMENTS:

Reviewer #1 (Remarks to the Author):

The authors have responded to all concerns raised by this and the other reviewers. The manuscript is now suitable for publication.

Reviewer #2 (Remarks to the Author):

The authors did significantly improve the manuscript, I do not have further comments.

Reviewer #3 (Remarks to the Author):

The authors have responded adequately to all my comments and suggestions. However, I still notice that the level of English is still not completely suitable yet for publication. This applies especially to the new text, but also, as I noticed before, to the original text (which does not seem to have much improved, if at all).

Please consider the following changes:

- Line 167 remove the word "almost"
- Lines 186-7: "...previous report showing that epidermal IgG deposition was markedly increased in inflamed-ears compared to untreated-ears⁶, we concluded that the transcellular IgG-extravasation pathway is less important under inflammatory conditions when paracellular leakage is dominant."
- Lines 195-198: Consider "IgG in HDBECs colocalized with EEA1, a marker of early endosomes, in foci within the cytoplasm (stained with CellMask Green), while being resistant to the acid wash by acetic acid³⁴, suggesting the IgG was internalized in HDBECs (Fig. 3A and Supplementary Fig. 2)."

Each text was revised exactly as the reviewer suggested.